# Yeast Ded1 promotes 48S translation pre-initiation complex assembly in an mRNA-specific and eIF4F-dependent manner

**Neha Gupta, Jon R Lorsch*, Alan G Hinnebusch***

Eunice Kennedy Shriver National Institute of Child Health and Human Development, National Institutes of Health, Bethesda, United States

**Abstract** DEAD-box RNA helicase Ded1 is thought to resolve secondary structures in mRNA 5'-untranslated regions (5'-UTRs) that impede 48S preinitiation complex (PIC) formation at the initiation codon. We reconstituted Ded1 acceleration of 48S PIC assembly on native mRNAs in a pure system, and recapitulated increased Ded1-dependence of mRNAs that are Ded1-hyperdependent in vivo. Stem-loop (SL) structures in 5'-UTRs of native and synthetic mRNAs increased the Ded1 requirement to overcome their intrinsically low rates of 48S PIC recruitment. Ded1 acceleration of 48S assembly was greater in the presence of eIF4F, and domains mediating one or more Ded1 interactions with eIF4G or helicase eIF4A were required for efficient recruitment of all mRNAs; however, the relative importance of particular Ded1 and eIF4G domains were distinct for each mRNA. Our results account for the Ded1 hyper-dependence of mRNAs with structure-prone 5'-UTRs, and implicate an eIF4E·eIF4G·eIF4A·Ded1 complex in accelerating 48S PIC assembly on native mRNAs.

DOI: https://doi.org/10.7554/eLife.38892.001

*For correspondence:
jon.lorsch@nih.gov (JRL);
ahinnebusch@nih.gov (AGH)

Competing interest: See
page 23

Reviewing editor: Nahum
Sonenberg, McGill University,
Canada

## Introduction

In canonical translation initiation in eukaryotes, a ternary complex (TC), consisting of eukaryotic initiation factor 2 (eIF2), Met-tRNA$_i$^Met, and GTP, along with eIF1, eIF1A, eIF5, eIF4B, and eIF3, binds to the small (40S) ribosomal subunit to form a 43S pre-initiation complex (PIC). The 43S PIC binds to the 5'-end of mRNA and scans the 5'-untranslated region (UTR) to identify the start codon, resulting in the formation of the 48S PIC. eIF4F complex, comprised of eIF4E (a cap binding protein), eIF4G (a scaffolding protein), and eIF4A (a DEAD-box RNA helicase), interacts with the mRNA m$^7$G cap and aids in recruitment of the 43S PIC to the 5'-end of the mRNA (reviewed in *Dever et al., 2016*; *Hinnebusch, 2014*).

eIF4A promotes 48S PIC formation in vitro and translation in vivo of virtually all mRNAs regardless of their structural complexity (*Pestova and Kolupaeva, 2002*; *Sen et al., 2015*; *Yourik et al., 2017*). Yeast ribosome profiling studies show that the majority of cellular mRNAs have strong and similar dependence on eIF4A for their proper translation in cells (*Sen et al., 2015*). Additionally, eIF4A is an essential protein, and small decreases in eIF4A cellular concentrations reduce bulk translation in vivo, further emphasizing the critical role of eIF4A in translation of most mRNAs (*Firczuk et al., 2013*). Yeast eIF4A is a weak helicase when unwinding RNA duplexes in vitro (*Rajagopal et al., 2012*; *Rogers et al., 1999*). Recent evidence suggests that mammalian eIF4A modulates the structure of the 40S subunit to enhance PIC attachment (*Sokabe and Fraser, 2017*).

Ded1 is a yeast DEAD-box RNA helicase that promotes translation in vivo of reporter mRNAs with longer or structured 5'-UTRs (*Berthelot et al., 2004*; *Chiu et al., 2010*; *Sen et al., 2015*). Like eIF4A, Ded1 is essential for yeast growth and stimulates bulk translation initiation in vivo (*Chuang et al., 1997*; *de la Cruz et al., 1997*). However, ribosome profiling of conditional *ded1*

mutants revealed that native mRNAs with 5'-UTRs that are longer and more structured than average yeast 5'-UTRs exhibit a greater than average reduction in translational efficiency (TE) relative to all other mRNAs on Ded1 inactivation (Ded1-hyperdependent mRNAs); whereas mRNAs with shorter and less structured 5'-UTRs exhibit increased relative TEs in *ded1* cells (Ded1-hypodependent mRNAs) (*Sen et al., 2015*).

Yeast Ded1 can unwind model RNA duplexes and act as an RNA chaperone or RNA-protein complex remodeler in vitro (*Bowers et al., 2006*; *Iost et al., 1999*; *Yang and Jankowsky, 2006*). Translation stimulation by Ded1 requires its ATPase activity (*Hilliker et al., 2011*; *Iost et al., 1999*). Mutations in the Ded1 ATPase domain (Ded1-E307A and Ded1-R489A) impair ATP binding and hydrolysis, and these mutants have dominant-negative effects on translation both in cell extracts and in vivo, evoking stress granule formation in the latter (*Hilliker et al., 2011*).

Ded1 can physically interact individually with purified eIF4E, eIF4G, eIF4A, or Pab1 (poly(A) binding protein), and can bind simultaneously to eIF4A·eIF4E·eIF4G (eIF4F) or eIF4E·eIF4G, in an RNA-independent manner (*Gao et al., 2016*; *Hilliker et al., 2011*; *Senissar et al., 2014*). Ded1 also interacts with the eIF4F complex in yeast extracts, supporting the physiological relevance of these interactions (*Hilliker et al., 2011*; *Senissar et al., 2014*). According to a proposed model, Ded1-eIF4G-mRNA interaction is thought to repress translation by promoting accumulation of mRNPs in stress granules, whereas ATP hydrolysis by Ded1 moves the repressed mRNPs back into the translation cycle (*Hilliker et al., 2011*). However, the role of Ded1-eIF4F interactions in the stimulatory function of Ded1 in translation initiation remains to be elucidated. Ded1 interacts with eIF4A through its N-terminal domain (Ded1-NTD) and this interaction is required for eIF4A stimulation of Ded1's RNA-duplex unwinding activity (*Gao et al., 2016*). Deletion of the Ded1-NTD confers a cold-sensitive growth phenotype in cells, consistent with a role for eIF4A stimulation of Ded1 function in vivo (*Banroques et al., 2011*; *Gao et al., 2016*). Yeast eIF4G contains three RNA binding domains, N-terminal RNA1, central RNA2, and C-terminal RNA3 (*Berset et al., 2003*); and while none of the three is essential, simultaneous deletion of RNA2 and RNA3 is lethal (*Park et al., 2011*). In vitro, eIF4G variants lacking any of the three RNA binding domains exhibit similar affinities for eIF4A, support similar rates of ATP-hydrolysis by the eIF4F complex (albeit with higher $K_m$ values for ATP), but lack the preference of WT eIF4F for 5'-overhang substrates during unwinding (*Rajagopal et al., 2012*). The Ded1 C-terminal domain (Ded1-CTD) interacts with the eIF4G-RNA3 domain, and Ded1-eIF4G interaction decreases the rate of RNA unwinding while increasing Ded1 affinity for RNA in vitro (*Hilliker et al., 2011*; *Putnam et al., 2015*). A Ded1 variant lacking the CTD conferred reduced reporter mRNA translation compared to WT Ded1 in cell extracts, supporting a stimulatory role for the Ded1-CTD interaction with eIF4G in translation initiation (*Hilliker et al., 2011*). Ded1 also interacts with the RNA2 domain of eIF4G and with eIF4E (*Senissar et al., 2014*), but the physiological relevance of these interactions is unknown.

We previously demonstrated functions of eIF4F, eIF4B, and eIF3 in stimulating the rate and extent of mRNA recruitment by 43S PICs in a fully purified yeast initiation system for native *RPL41A* mRNA, containing a short and relatively unstructured 5'-UTR (*Mitchell et al., 2010*). Although Ded1 is essential in vivo, it was dispensable for recruitment of this mRNA in vitro. Considering that *RPL41A* was judged to be Ded1-hypodependent in vivo by ribosome profiling of *ded1* mutants (*Sen et al., 2015*), we asked whether recruitment of Ded1-hyperdependent mRNAs would require Ded1 in the reconstituted system. We investigated whether the presence of defined stem-loop (SL) structures in native or synthetic mRNAs would confer greater Ded1-dependence for rapid recruitment in vitro. Finally, we examined the role of the RNA2 and RNA3 domains of eIF4G and the NTD and CTD of Ded1 that mediate Ded1 interactions with the eIF4F complex in promoting Ded1's ability to accelerate mRNA recruitment by 43S PICs. Our findings demonstrate that Ded1 accelerates recruitment of native and synthetic mRNAs, overcoming the inhibitory effects of structured leader sequences and conferring relatively greater stimulation for mRNAs hyperdependent on Ded1 in vivo, all in a manner consistent with stimulation of Ded1 function by formation of a Ded1-eIF4F complex.

## Results

### Ded1 enhances the rate of recruitment of all natural mRNAs tested

We set out to reconstitute the function of Ded1 in 48S PIC assembly in a yeast translation initiation system comprised of purified components (*Mitchell et al., 2010*; *Walker et al., 2013*; *Yourik et al., 2017*). Pre-assembled 43S PICs, containing 40S subunits and factors eIF1, eIF1A, eIF5, eIF2·GDPNP·Met-tRNA$_i$, eIF4G·4E, eIF4A, eIF4B, and eIF3, were pre-incubated with or without Ded1, and reactions were initiated by addition of ATP and $^{32}$P-labeled m$^7$Gppp-capped mRNA (synthesized in vitro). Formation of 48S complexes was monitored over time using a native gel electrophoretic mobility shift assay (EMSA) to resolve free and 48S-bound mRNAs. An ~20 fold excess of unlabeled-capped mRNA was added to reaction aliquots at each time point to quench further recruitment of $^{32}$P-labeled mRNA ('pulse-quench'). By varying the concentration of Ded1, this assay yields the apparent rate constants (k$_{app}$) for 48S PIC formation at each Ded1 concentration, the maximal rate at saturating Ded1 (k$_{max}$), the Ded1 concentration required for the half-maximal rate of 48S formation (K$_{1/2}$), and the reaction endpoints (percentage of mRNA recruited) at each Ded1 concentration. The addition of ~20 fold excess non-radiolabeled mRNA in the quench was adequate to prevent further recruitment of $^{32}$P-labeled mRNA, and did not dissociate the pre-formed $^{32}$P-labeled 48S complexes on the timescale of the recruitment experiments (*Figure 1—figure supplement 1A–B* and *Figure 2—figure supplement 1E*). Our purified Ded1 hydrolyzed ATP in an RNA-dependent manner with k$_{cat}$ and $\mathrm{K}_m^{\mathrm{ATP}}$ values consistent with previous measurements (*Figure 1—figure supplement 1C–E*) (*Iost et al., 1999*; *Senissar et al., 2014*). Moreover, the Ded1 bound a fluorescently labeled single-stranded mRNA in the presence or absence of ADP or ADPNP with K$_D$ values consistent with published values (*Figure 1—figure supplement 1F*) (*Banroques et al., 2008*; *Iost et al., 1999*).

We began by analyzing the effect of Ded1 on recruitment of *RPL41A* mRNA, a short transcript of 310 nucleotides (nt), with 5'-UTR of only 24 nt (*Figure 1A* and *Figure 1—figure supplement 1G*), and a low degree of predicted secondary structure (*Mitchell et al., 2010*). *RPL41A* behaved like a Ded1-hypodependent mRNA in ribosome profiling experiments, exhibiting increased relative TE in *ded1* mutant versus WT cells (*Sen et al., 2015*). Consistent with this, recruitment of *RPL41A* mRNA in vitro was achieved previously without Ded1 (*Mitchell et al., 2010*). Nevertheless, with our more sensitive pulse-quench approach, we found that addition of saturating Ded1 increased the k$_{max}$ of *RPL41A* by ~2.8 fold from 0.95 ± 0.1 min$^{-1}$ to 2.7 ± 0.3 min$^{-1}$ (*Figure 1B*, cf. blue vs. orange, and *Figure 1—figure supplement 1H*-red), with a $\mathrm{K}_{1/2}^{\mathrm{Ded1}}$ of 58 ± 8 nM (*Figure 1C*). Thus, despite its low Ded1-dependence relative to most other mRNAs in vivo, *RPL41A* recruitment is appreciably stimulated by Ded1 in vitro.

We next examined Ded1 stimulation of another Ded1-hypodependent mRNA, *HOR7*, and several Ded1-hyperdependent mRNAs, *SFT2*, *PMA1*, *OST3*, and *FET3*, which exhibited reduced TEs in *ded1* versus WT cells (*Sen et al., 2015*). *SFT2* and *PMA1* mRNAs are noteworthy in containing SL structures in their 5'-UTRs detected in vivo (*Rouskin et al., 2014*). Because these mRNAs exceed the maximum length that can be resolved by EMSA (~400 nt), we generated reporter mRNAs with the 5'-UTR and first 60 nt of coding sequences (CDS) of each mRNA (*Figure 1A* and *Figure 1—figure supplement 1G*). (For brevity, we refer to these reporter mRNAs simply by their gene names.) Without Ded1, the rate of *HOR7* recruitment was 1.3 ± 0.16 min$^{-1}$, and saturating Ded1 stimulated recruitment by ~2 fold (k$_{max}$ = 2.8 ± 0.4 min$^{-1}$) with a $\mathrm{K}_{1/2}^{\mathrm{Ded1}}$ of 115.5 ± 17 nM, results that were similar to those for Ded1-hypodependent *RPL41A* (*Figure 1B–C* and *Figure 1—figure supplement 1H*-blue curve). Strikingly, without Ded1, the four Ded1-hyperdependent mRNAs were recruited at rates 2.5- to 15-fold lower compared to the two Ded1-hypodependent mRNAs (*Figure 1B*, blue), and these rates increased by an order of magnitude on addition of Ded1 (*Figure 1B*, orange and bottom panel; and *Figure 1—figure supplement 1H*). The Ded1-hyperdependent mRNAs also required higher Ded1 concentrations to achieve these maximal rates (*Figure 1C*). The acceleration of mRNA recruitment by Ded1 required its ATPase activity, as ATPase-deficient Ded1 variant Ded1$^{E307A}$ (*Figure 1—figure supplement 1D–E*) (*Iost et al., 1999*) did not accelerate recruitment of any of the mRNAs (*Figure 1—figure supplement 1I*). Moreover, the presence of Ded1$^{E307A}$ interfered with stimulation of *RPL41A* recruitment by WT Ded1, increasing the $\mathrm{K}_{1/2}^{\mathrm{Ded1}}$ by ~4 fold (*Figure 1—figure supplement 1J*).

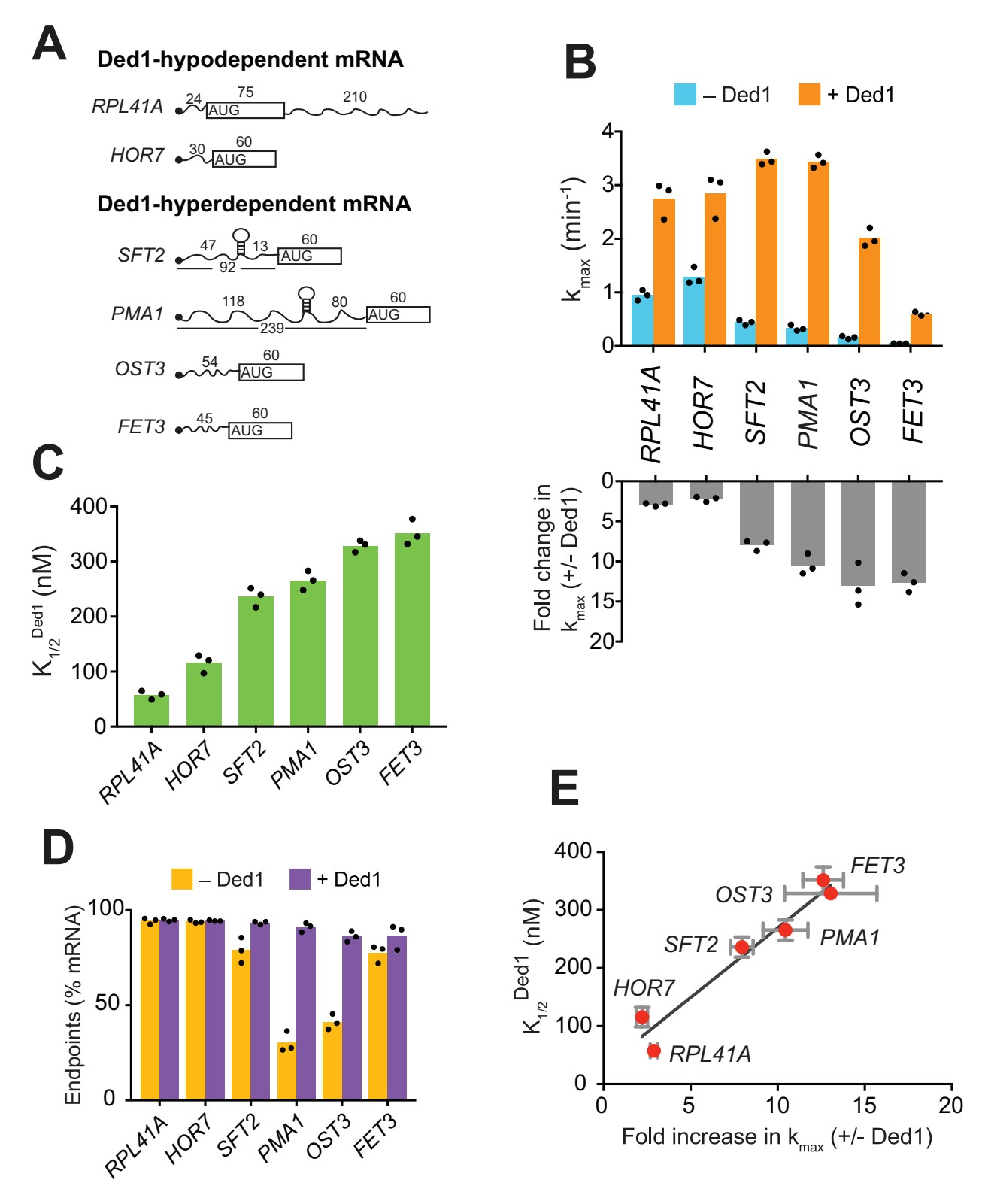

**Figure 1.** Ded1 confers relatively greater acceleration of 48S PIC assembly in vitro for native mRNAs hyperdependent on Ded1 in vivo. (**A**) Schematics of native *RPL41A* mRNA, and mRNA reporters for other native yeast mRNAs comprised of the 5'-UTR and first 60 nt of the ORF. ORFs are depicted as boxes with an AUG start codon; wavy lines depict 5'- and 3'-UTRs of the indicated nucleotide lengths; black balls depict m⁷Gppp caps; SLs in the 5'-UTRs of *SFT2* and *PMA1* identified in vivo are shown as hairpins, whose folding free energies are given in *Figure 2—figure supplement 1A*. (**B**) *Upper:*
*Figure 1 continued on next page*

*Figure 1 continued*

Maximum rates of recruitment ($k_{max}$) in the absence (blue) and presence (orange) of saturating Ded1 for mRNAs in (A). *Lower*: Fold-changes in $k_{max}$ with or without Ded1 ($k_{max}^{+Ded1}/k_{max}^{-Ded1}$) calculated from data in upper plot. (C) Concentrations of Ded1 resulting in half-maximal rates ($K_{1/2}^{Ded1}$). (D) Endpoints of mRNA recruitment as percentages of total mRNA (15 nM) bound to 48S PICs (30 nM) in absence (gold) or presence (purple) of saturating Ded1. (E) Plot of $K_{1/2}^{Ded1}$ from (C) versus fold-change in $k_{max}^{+Ded1}/k_{max}^{-Ded1}$ from (B, lower) for each mRNA. Red points and error bars indicate mean values and one standard deviation, respectively, for each parameter. Line generated by linear regression analysis (Y = 24.02*X + 28.98, $R^2$ = 0.944, p-value=0.001). (B–D) Bars indicate mean values calculated from the three independent experiments represented by the black data points. See *Figure 1—figure supplement 1* and *Figure 1 —source data 1*.

DOI: https://doi.org/10.7554/eLife.38892.002

The following source data and figure supplements are available for figure 1:

**Source data 1.** (Source data file for *Figure 1*).
DOI: https://doi.org/10.7554/eLife.38892.005
**Figure supplement 1.** Catalytically active Ded1 stimulates 48S PIC assembly on native mRNAs.
DOI: https://doi.org/10.7554/eLife.38892.003
**Figure supplement 1—source data 1.** (Source data file for *Figure 1—figure supplement 1*).
DOI: https://doi.org/10.7554/eLife.38892.004

*PMA1* and *OST3* also exhibited low endpoints of recruitment without Ded1, 30 ± 5% and 41 ± 4%, respectively, which increased to 91 ± 2% and 86 ± 3%, respectively, on Ded1 addition (*Figure 1D*). It was suggested that Ded1 acts as an RNA chaperone to aid transitions between different RNA conformations (*Yang et al., 2007*). In fact, two or more conformers of *OST3* (and other mRNAs) were observed in native gel electrophoresis that likely represent differently folded, stable mRNA conformers (*Figure 1—figure supplement 1K*). Perhaps only one of these conformers of *PMA1* and *OST3* is competent for 48S PIC assembly, and Ded1 facilitates isomerization among them.

Interestingly, a linear relationship was observed between the fold-acceleration by Ded1 and the $K_{1/2}^{Ded1}$, such that Ded1-hypodependent and Ded1-hyperdependent mRNAs cluster separately from each other along the line (*Figure 1E*). One explanation could be that Ded1-hyperdependent mRNAs have a higher rate-limiting activation energy barrier for 48S PIC assembly in the absence of Ded1 compared to the Ded1-hypodependent mRNAs, consistent with the latter's relatively higher rates of recruitment without Ded1 (*Figure 1B*, blue bars, *RPL41A* and *HOR7* versus *SFT2*, *PMA1*, *OST3*, and *FET3*). Accordingly, the hyperdependent mRNAs require relatively higher Ded1 concentrations to lower this barrier to the point at which a Ded1-independent step becomes rate limiting (*SFT2* and *PMA1*) or where Ded1 cannot lower the Ded1-dependent barrier further (*OST3* and *FET3*) (*Figure 1—figure supplement 1L*).

Taken together, the data presented above demonstrate that mRNAs with long and structured 5'-UTRs, found to be hyperdependent on Ded1 for translation in vivo, are inherently less capable of PIC recruitment and more dependent on Ded1 for rapid recruitment in vitro than are mRNAs hypodependent on Ded1 in vivo.

## Secondary structures in the 5'-UTR increase Ded1-dependence in 48S PIC assembly

Given that mRNAs with heightened Ded1-dependence in vivo have a greater than average potential to adopt secondary structures involving the 5'-UTR (*Sen et al., 2015*), we investigated if the stable SL structures previously detected in the 5'-UTRs of *SFT2* and *PMA1* were responsible for their elevated Ded1 dependence (*Rouskin et al., 2014*). To this end, we introduced mutations to eliminate the SL in each mRNA, or (for *SFT2*) to strengthen the SL (*Figure 2A* and *Figure 2—figure supplement 1A–B*). Without Ded1, the SL-disrupted version of *PMA1*, *PMA1-M*, showed ~2 fold higher endpoints of recruitment than WT *PMA1* (*PMA1-M* = 62 ± 2.5%, *PMA1* = 30 ± 5%, *Figure 2B*, gold). *PMA1-M* was also recruited at rates ~ 4 fold higher than *PMA1* ($k_{max}$ = 1.2 ± 0.1 min$^{-1}$ (*PMA1-M*) vs. 0.33 ± 0.06 min$^{-1}$ (*PMA1*); *Figure 2C*, blue), consistent with the idea that the SL inhibits mRNA recruitment. Ded1 increased the rate of *PMA1-M* recruitment to yield a $k_{max}$ similar to that for *PMA1* (*Figure 2C*, orange), but at a much lower Ded1 concentration for *PMA1-M* ($K_{1/2}^{Ded1}$ = 40 ± 6 nM) versus *PMA1* (~280 nM) (*Figure 2D* and *Figure 2—figure supplement 1C*, black vs. purple).

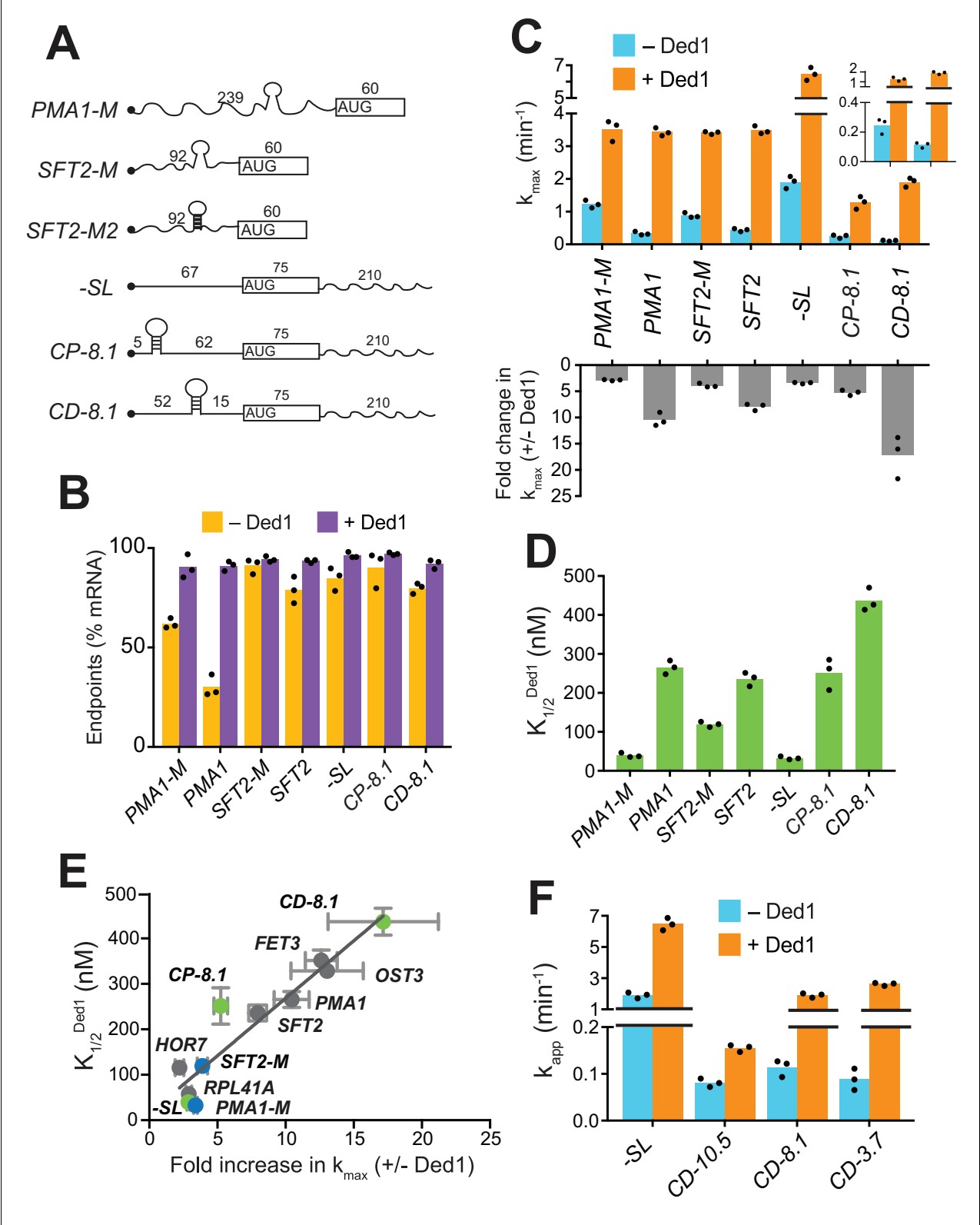

**Figure 2.** Secondary structures in 5'-UTRs confer Ded1-hyperdependence in accelerating 48S PIC assembly in vitro. (**A**) Schematics of derivatives of natural mRNAs from *Figure 1A* mutated to decrease (*SFT2-M* and *PMA1-M*) or increase (*SFT2-M2*) the stability of 5'-UTR SLs; and synthetic mRNAs depicted as in *Figure 1A*. The sequences of SLs and mutations are in *Figure 2—figure supplement 1A*. (**B**) Endpoints of 48S PIC assembly reactions, determined as in *Figure 1D*. (**C**) $k_{max}$ values (upper) and fold-change in $k_{max}$ (lower) in the presence versus absence of Ded1, determined as in *Figure 2 continued on next page*

Figure 2 continued

**Figure 1B**. The inset in the upper panel scales the y-axis to display the low $k_{max}$ values for *CD-8.1* and *CP-8.1* mRNAs without Ded1. Note that Y-axis is discontinuous. (D) $K_{1/2}^{Ded1}$ values determined as in **Figure 1C**. (E) Plot of $K_{1/2}^{Ded1}$ from (D) versus fold-change in $k_{max}^{+Ded1}/k_{max}^{-Ded1}$ from (C, lower) for the indicated mRNAs. Gray, blue, and green points indicate natural, mutated, and synthetic mRNAs, respectively, with error bars indicating 1 SD; line produced by linear regression analysis (Y = 25.29*X + 15.28, $R^2$ = 0.89, p-value<0.001). (F) Apparent rate of mRNA recruitment of synthetic mRNAs with different strengths SLs in the cap-distal region in the absence (blue) and presence of Ded1 (orange). ΔG°: *CD-10.5* = −10.5 kcal/mol, *CD-8.1* = −8.1 kcal/mol, and *CD-3.7* = −3.7 kcal/mol. Note that Y-axis is discontinuous. (B–D, F) Bars indicate mean values calculated from the three independent experiments represented by the data points. (B–D) WT *PMA1* amd WT *SFT2* data is added from **Figure 1** for comparison. See **Figure 2—figure supplement 1** and **Figure 2—source data 1**.

DOI: https://doi.org/10.7554/eLife.38892.006

The following source data and figure supplements are available for figure 2:

**Source data 1.** (Source data file for **Figure 2**).

DOI: https://doi.org/10.7554/eLife.38892.008

**Figure supplement 1.** Ded1 enhances the recruitment of mRNAs with SLs in the 5'-UTRs.

DOI: https://doi.org/10.7554/eLife.38892.007

**Figure supplement 1—source data 1.** (Source data file for **Figure 2—figure supplement 1**).

DOI: https://doi.org/10.7554/eLife.38892.009

Similar results were observed for the SL-disrupted *SFT2* variant, *SFT2-M*, as follows. Compared to WT *SFT2* mRNA, *SFT2-M* showed an ~2 fold faster recruitment in the absence of Ded1 (0.88 ± 0.08 min⁻¹ vs. 0.44 ± 0.05 min⁻¹, **Figure 2C**, blue) and an ~2 fold lower Ded1 concentration required to accelerate recruitment to half the maximum rate: $K_{1/2}^{Ded1}$ = 119 ±7 nM (*SFT2-M*) vs. 236±18 nM (*SFT2*) (**Figure 2C**, orange; **Figure 2D** and **Figure 2—figure supplement 1C**, green vs. orange curves). Importantly, both *PMA1-M* and *SFT2-M* mRNAs clustered with the Ded1-hypodependent mRNAs instead of the Ded1-hyperdependent mRNAs in the plot of fold-acceleration versus $K_{1/2}^{Ded1}$ (**Figure 2E**, blue). In contrast, the *SFT2-M2* variant harboring a SL of enhanced stability (ΔG° = −19.1 kcal/mol vs. −9.4 kcal/mol for WT *SFT2*; **Figure 2A** and **Figure 2—figure supplement 1A–B**) was not recruited without Ded1, and exhibited only low-level recruitment at the highest achievable Ded1 concentration (**Figure 2—figure supplement 1D**). These results strongly suggest that Ded1 accelerates the recruitment of WT *SFT2* and *PMA1* mRNAs, and increases the endpoint of *PMA1* recruitment, in part by melting the SL structures in their 5'-UTRs. As the *PMA1* SL is ~120 nt from the 5'-cap, it is likely that Ded1 resolves the SL to accelerate scanning of the PIC through the 5'-UTR.

## Evidence that Ded1 stimulates the PIC attachment and scanning steps of initiation

Although Ded1-hypodependent mRNAs *RPL41A* and *HOR7* lack any strong, defined SLs, Ded1 still stimulated their recruitment (**Figure 1B**). Similarly, even after removal of SLs, the *PMA1-M* and *SFT2-M* mutant mRNAs were still stimulated by Ded1 (**Figure 2C**). In addition to defined, stable SLs in 5'-UTRs, natural mRNAs likely form large ensembles of weaker structures involving interactions of nucleotides within the 5'-UTR or between 5'-UTR nucleotides and nucleotides in the CDS or 3'-UTR (**Kertesz et al., 2010**; **Yourik et al., 2017**), which might also contribute to Ded1-dependence during PIC attachment or scanning. To test this hypothesis, we examined recruitment of chimeric mRNAs with synthetic 5'-UTRs attached to the CDS and 3'-UTR of native *RPL41A* (**Yourik et al., 2017**). A synthetic mRNA dubbed '-*SL*' (for 'minus stem-loop') contained a 67 nt 5'-UTR comprised of CAA repeats that is devoid of stable secondary structure (**Figure 2A** and **Figure 2—figure supplement 1B**). Without Ded1, -*SL* was recruited rapidly at a rate of 1.9 ± 0.2 min⁻¹ (**Figure 2C**, blue),~2 fold faster than *RPL41A* (**Figure 1B**, blue) or the *SFT2-M* and *PMA1-M* variants lacking SLs (**Figure 2C**, blue), consistent with the idea that weaker interactions formed by 5'-UTRs of native mRNAs lacking SLs can inhibit recruitment. Ded1 increased the $k_{max}$ of -*SL* recruitment by ~3 fold (**Figure 2C**, orange; and **Figure 2—figure supplement 1F**, red), comparable to the increases observed for *SFT2-M*, *PMA1-M* (**Figure 2C**, lower), and *RPL41A* (**Figure 1B**, lower). This stimulation of -*SL* recruitment was achieved at a relatively low Ded1 concentration comparable to, or lower than, those observed for *SFT2-M*, *PMA1-M* (**Figure 2D**), and *RPL41A* (**Figure 1C**). Hence, -*SL* mRNA clustered with the Ded1-hypodependent mRNAs shown in the plot of $K_{1/2}^{Ded1}$ versus fold-increase in $k_{max}$ (**Figure 2E**,

green, *-SL*). The finding that Ded1 significantly stimulates 48S PIC assembly on *-SL*, containing a 5'-UTR incapable of forming stable structures on its own, suggests that Ded1 has a second role in mRNA recruitment, in addition to unwinding stable structures in 5'-UTRs.

To analyze whether Ded1 can stimulate the PIC attachment step of 48S assembly, we examined the synthetic *CP-8.1* mRNA with a stable SL (predicted ΔG° = −8.1 kcal/mol) inserted in a cap-proximal location 5 nt from the 5'-end of *-SL* mRNA. We also analyzed a third synthetic mRNA, *CD-8.1*, containing the same SL inserted 45 nt downstream from the cap of *-SL*, reasoning that this cap-distal SL might impede scanning without interfering with PIC attachment at the cap (*Figure 2A* and *Figure 2—figure supplement 1A–B*). As expected, both synthetic mRNAs with SLs had order-of-magnitude lower rates of recruitment compared to *-SL* in the absence of Ded1 (*Figure 2C*, blue, inset), indicating that SLs in either location strongly inhibited 48S PIC formation. Ded1 increased the maximal rate of *CP-8.1* recruitment by ~5 fold, such that the maximal rate for *CP-8.1* was still ~6 fold below that of *-SL* (*Figure 2C* and *Figure 2—figure supplement 1F*, blue vs. red), and ~2 − 3 fold below that of *RPL41A, HOR7, SFT2, OST3*, and *PMA1* (*Figure 2C* and *Figure 1B*). Moreover, *CP-8.1* exhibited an ~7.6 fold higher $\mathrm{K}_{1/2}^{\mathrm{Ded1}}$ versus *-SL* (*Figure 2D*). As a result, *CP-8.1* lies between the Ded1-hyperdependent and Ded1-hypodependent mRNAs, and deviates from the line in the plot of $\mathrm{K}_{1/2}^{\mathrm{Ded1}}$ versus fold-increase in $k_{max}$ (*Figure 2E*, green, *CP-8.1*). This deviation is due to the fact that *CP-8.1*, like the Ded1-hyperdependent mRNAs, is inefficiently recruited in the absence of Ded1; but unlike the Ded1-hyperdependent mRNAs, its recruitment is accelerated by Ded1 only ~5 fold. These results suggest that Ded1 is not very effective at reducing the inhibitory effect of a stable cap-proximal SL on PIC attachment, even at saturating Ded1 concentrations. In contrast, Ded1 conferred an ~17 fold acceleration of recruitment for *CD-8.1* (*Figure 2C*), and a high Ded1 concentration was required to achieve the half-maximal rate (*Figure 2D* and *Figure 2—figure supplement 1F*, green vs. red), similar to the behavior of the native Ded1-hyperdependent mRNAs (*Figure 1B–C*). Judging by the relative positions of *CD-8.1* and *CP-8.1* on the plot of *Figure 2E*, it appears that Ded1 is better at resolving the inhibitory effect of the synthetic SL in a cap-distal versus cap-proximal location in the 5'-UTR. Interestingly, the same conclusion was reached previously from analyzing the relative effects of a cold-sensitive *ded1* mutation on expression of reporter mRNAs containing cap-proximal or cap-distal SLs (*Sen et al., 2015*). The ATPase deficient mutant Ded1^E307A did not increase the recruitment rates for these mRNAs above the levels seen in the absence of Ded1, even in case of *-SL* mRNA which lacks stable secondary structures in the 5'-UTR (*Figure 2—figure supplement 1G*).

Having observed that the strength of a 5'-UTR SL influences the degree of Ded1-dependence, as observed with WT versus mutant derivatives of *SFT2* and *PMA1* (*Figure 2C*), we went on to analyze synthetic mRNAs containing cap-distal SLs of predicted stabilities either higher (−10.5 kcal/mol) or lower (−3.7 kcal/mol) than that of *CD-8.1*. Each of these SLs, present in *CD-10.5* and *CD-3.7*, respectively, reduced the recruitment rate in the absence of Ded1 by ~20 fold compared to that of *-SL*, similar to the results for *CD-8.1* (*Figure 2F*, blue). This result suggests that eIF4E·eIF4G·eIF4A alone cannot efficiently mitigate the inhibitory effects of cap-distal SLs of even moderate stability, such as that in *CD-3.7*, consistent with previous findings (*Yourik et al., 2017*). As expected, Ded1 strongly stimulated the recruitment rate of *CD-3.7*, by ~25 fold, slightly more than the ~15 fold observed for *CD-8.1* (*Figure 2F*); however, Ded1 conferred only a modest ~2 fold acceleration of *CD-10.5* recruitment (*Figure 2F*). One possibility to explain these last results would be that the cap-distal SL in *CD-10.5* is too stable for efficient unwinding by Ded1, limiting Ded1's ability to accelerate 48S assembly on this mRNA.

In summary, our analysis of synthetic mRNAs supports the notion that Ded1 can accelerate mRNA recruitment by enhancing scanning through cap-distal secondary structures, such as the SLs that occur naturally in *SFT2* and *PMA1* mRNAs; although if the structure is too stable, Ded1 has a limited ability to unwind it. Additionally, it appears that Ded1 also partially mitigates the inhibitory effects of cap-proximal secondary structures on PIC attachment at the mRNA 5'-end, although not to the same degree that it overcomes cap-distal structures.

## Ded1 stimulation is completely dependent on eIF4E·eIF4G for a subset of mRNAs and all mRNAs require eIF4A in the presence or absence of Ded1

Ded1 has been shown to interact physically with eIF4F components eIF4E, eIF4G and eIF4A in a manner that influences its ability to unwind model RNA substrates in an unwinding assay (*Gao et al., 2016*; *Hilliker et al., 2011*; *Senissar et al., 2014*). Accordingly, we examined whether its interactions with eIF4F influence Ded1's ability to accelerate 48S PIC assembly. As eIF4E is co-purified with eIF4G (*Mitchell et al., 2010*), and eIF4E is required for full activity of the eIF4F complex, all experiments involving eIF4G utilize the eIF4E·eIF4G heterodimer. We performed mRNA recruitment assays in the presence and absence of eIF4E·eIF4G and Ded1 for (i) Ded1-hypodependent mRNAs *RPL41A*, *HOR7*, *-SL*, and *SFT2-M*; (ii) Ded1-hyperdependent mRNAs *SFT2*, *OST3*, and *CD-8.1*; and (iii) *CP-8.1*, which exhibits intermediate behavior between groups (i) and (ii) mRNAs in the relationship between $K_{1/2}^{Ded1}$ and $k_{max}$ stimulation (*Figure 2E*). The 5'-cap blocks aberrant mRNA recruitment and imposes a requirement for eIF4E·eIF4G and eIF4A for maximal recruitment rate. 43S PICs can bind to uncapped mRNAs but are unable to either locate or stably associate with their start codons (*Mitchell et al., 2010*). Accordingly, only 5'-capped mRNAs were used in all experiments to avoid the formation of these aberrant complexes, especially in the absence of eIF4E·eIF4G or eIF4A.

As described above, in the presence of eIF4E·eIF4G, all eight mRNAs can be recruited in the absence of Ded1, and Ded1 increased their recruitment rates to different extents (*Figure 3A–H*, blue vs. orange). *RPL41A*, *HOR7*, and *-SL*, which contain very short (*RPL41A* and *HOR7*) or unstructured (*-SL*) 5'-UTRs, were recruited at relatively low (but measurable) rates in the absence of both eIF4E·eIF4G and Ded1 (*Figure 3A,B,F*, tan bars); and Ded1 conferred no increase in their apparent rates in the absence of eIF4E·eIF4G (*Figure 3A,B,F*, cf. green vs. tan, orange vs. blue). The complete dependence of Ded1 on eIF4E·eIF4G to accelerate recruitment for these three mRNAs is consistent with the idea that Ded1 acts exclusively in the context of the eIF4G·eIF4E·eIF4A·Ded1 quaternary complex (*Gao et al., 2016*). The same might be true for *CP-8.1*, whose observable stimulation by Ded1 also required eIF4E·eIF4G (*Figure 3H*); however, because no *CP-8.1* recruitment was observed without eIF4E·eIF4G, Ded1 might stimulate recruitment of this mRNA on its own at levels below the detection limit of the assay. The finding that *CP-8.1* differs from *-SL* in showing no measurable recruitment in the absence of eIF4E·eIF4G (*Figure 3H* vs 3F), suggests that the cap-proximal SL in *CP-8.1* imposes a requirement for eIF4E·eIF4G.

In contrast to the results described above, Ded1 can accelerate recruitment of *SFT2*, *OST3*, *SFT2-M*, and *CD-8.1* mRNAs in the absence of eIF4E·eIF4G, with comparable apparent rates afforded by eIF4E·eIF4G alone for *SFT2*, *OST3*, and *CD-8.1* (*Figure 3C,D,G*; green vs. blue), but well below the apparent rates observed with eIF4E·eIF4G for *SFT2-M* (*Figure 3E*, green vs. blue). Thus, Ded1 can stimulate recruitment of these four mRNAs, at least to some extent, acting outside of the eIF4G·eIF4E·eIF4A·Ded1 complex. However, because the maximal rates were observed in the presence of both eIF4E·eIF4G and Ded1 (*Figure 3C–E,G*, orange bars), Ded1 likely functions within the eIF4G·eIF4E·eIF4A·Ded1 complex as well for these four mRNAs.

In contrast to eIF4E·eIF4G, omitting eIF4A from the reactions essentially eliminated recruitment of all mRNAs tested, yielding endpoints of <10%; with the exception of *-SL*, which was recruited at the low rate of 0.04 ± 0.01 min⁻¹ with endpoints of 46 ± 3% (*Figure 3—figure supplement 1*, red). Moreover, in the absence of eIF4A, Ded1 did not rescue recruitment of any mRNAs, nor did it increase the $k_{app}$ or endpoints for *-SL* mRNA (*Figure 3—figure supplement 1*, blue). Thus, eIF4A has one or more essential functions in mRNA recruitment that cannot be provided by Ded1, even for an mRNA such as *-SL* exhibiting appreciable recruitment in the absence of eIF4E·eIF4G (*Figure 3F*, tan). This is consistent with the previous findings that eIF4A is required for robust translation of virtually all yeast mRNAs in vivo and in vitro (*Sen et al., 2015*; *Yourik et al., 2017*), and that DHX29 and yeast Ded1 cannot substitute for eIF4A in 48S PIC assembly on native β-globin mRNA in a mammalian reconstituted system (*Abaeva et al., 2011*; *Pisareva et al., 2008*). Moreover, the fact that Ded1 does not accelerate recruitment of *-SL* mRNA in the absence of eIF4A (*Figure 3—figure supplement 1H*, blue vs. red) indicates that, at least for this mRNA with an unstructured leader, Ded1 can only promote mRNA recruitment in the presence of eIF4A.

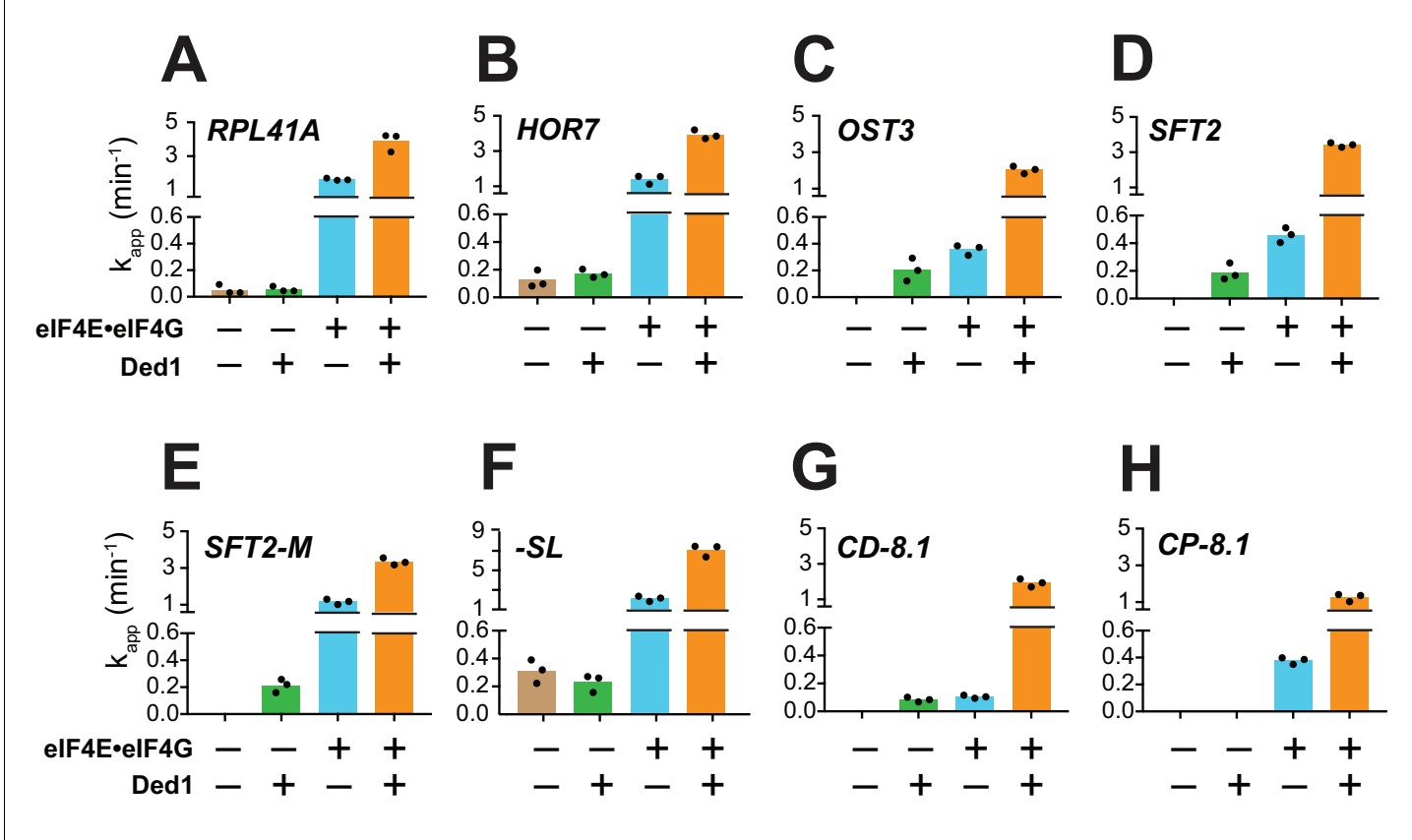

**Figure 3.** Ded1 acceleration of 48S PIC assembly is completely dependent on eIF4E·eIF4G for a subset of mRNAs. (**A–H**) Apparent rates ($k_{app}$) of 48S PIC assembly for the indicated mRNAs observed without eIF4E·eIF4G and Ded1 (tan bars), with Ded1 but no eIF4E·eIF4G (green bars), with eIF4E·eIF4G but no Ded1 (blue bars), or with both Ded1 and eIF4E·eIF4G (orange bars). Note the different Y-axis scale for *-SL* (**F**). See *Figure 3—figure supplement 1* and *Figure 3—source data 1*.

DOI: https://doi.org/10.7554/eLife.38892.010

The following source data and figure supplements are available for figure 3:

**Source data 1.** (Source data file for *Figure 3*).

DOI: https://doi.org/10.7554/eLife.38892.012

**Figure supplement 1.** eIF4A is required for appreciable recruitment of all mRNAs tested (except *-SL*) in the presence or absence of saturating Ded1.

DOI: https://doi.org/10.7554/eLife.38892.011

**Figure supplement 1—source data 1.** (Source data file for *Figure 3—figure supplement 1*).

DOI: https://doi.org/10.7554/eLife.38892.013

## Interaction between the RNA3 domain of eIF4G and Ded1-CTD stimulates mRNA recruitment

The C-terminal RNA3 domain of eIF4G was shown to interact physically with the Ded1 CTD (*Hilliker et al., 2011*) and to influence effects of eIF4G on Ded1 unwinding of a model RNA duplex in vitro (*Gao et al., 2016*; *Putnam et al., 2015*). Hence, we sought to determine whether this physical interaction between the CTDs of Ded1 and eIF4G is functionally relevant in 48S PIC assembly by performing mRNA recruitment assays with a truncated eIF4G variant lacking RNA3 (eIF4E·eIF4G-Δ RNA3) or a Ded1 variant lacking the CTD (Ded1-ΔCTD) (*Figure 4—figure supplement 1A*).

Without Ded1, the ΔRNA3 truncation of eIF4G had little or no effect on $k_{max}$ for all seven mRNAs tested (*Figure 4A*, compare blue bars to superimposed line/whiskers, the latter indicating results for WT eIF4E·eIF4G re-plotted from *Figure 3A–H* for comparison, where the horizontal line shows the mean and the whiskers one SD from the mean). In the presence of Ded1, by contrast, ΔRNA3 conferred ~2 fold reductions in $k_{max}$ for three mRNAs, *RPL41A* (ΔRNA3 – $1.5 \pm 0.2$ min$^{-1}$; WT – $3.9 \pm 0.5$ min$^{-1}$), *HOR7* (ΔRNA3 – $1.8 \pm 0.1$ min$^{-1}$; WT – $3.9 \pm 0.3$ min-1) and *CP-8.1* (ΔRNA3 – 0.7 ±

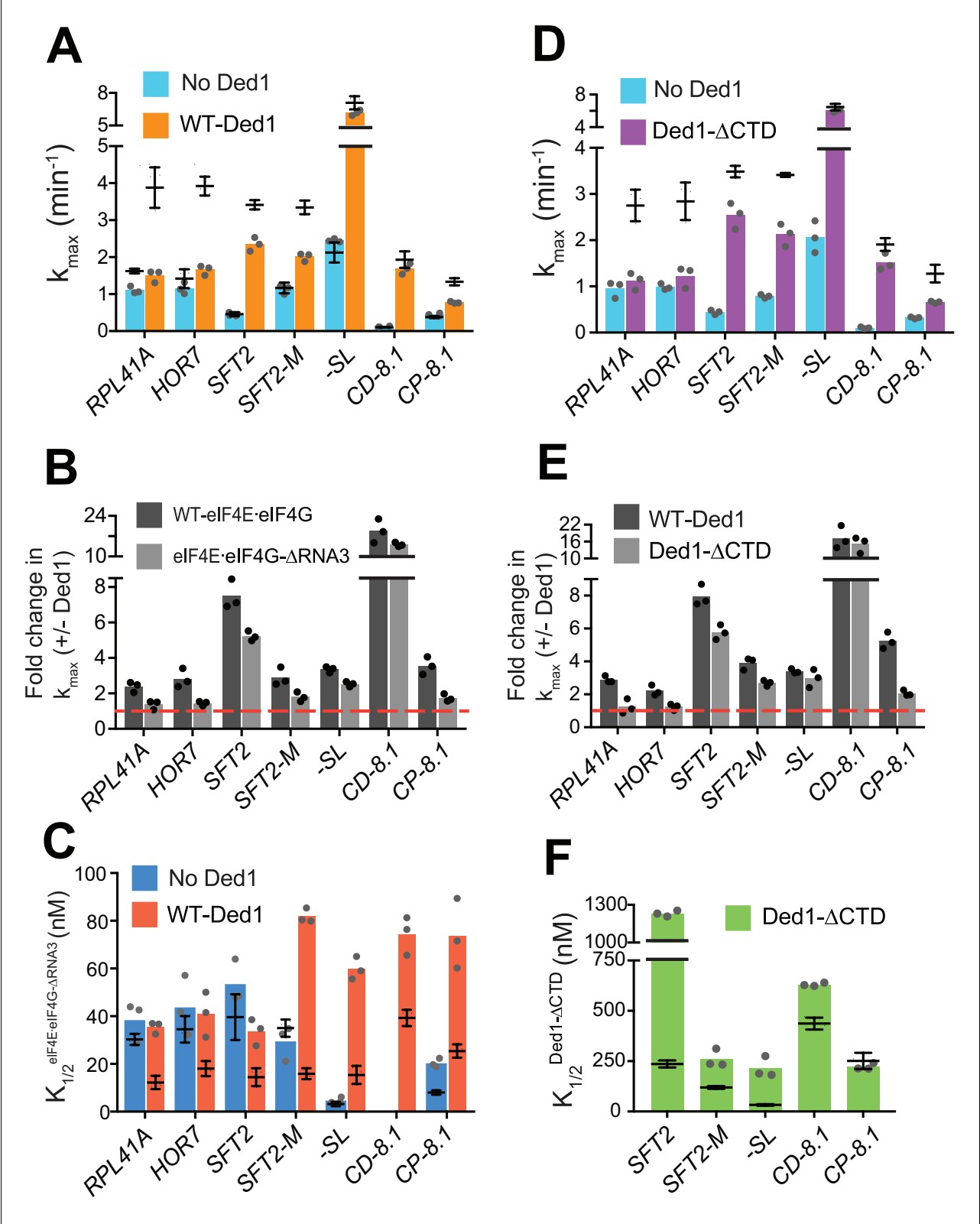

**Figure 4.** eIF4G-RNA3 and Ded1-CTD enhance $k_{max}$ for the same subset of mRNAs while reducing $K_{1/2}$ for nearly all mRNAs. (A) $k_{max}$ values in absence (blue) and presence of saturating Ded1 (orange) with eIF4E·eIF4G-ΔRNA3. (B) The average fold-change in the kmax observed in the presence and absence of Ded1 ($k_{max}^{+Ded1}/k_{max}^{-Ded1}$) with WT eIF4E·eIF4G (dark grey) and eIF4E·eIF4G-ΔRNA3 (light grey). Red line indicates no change in the rate on Ded1 addition. (C) $K_{1/2}$ of eIF4E·eIF4G-ΔRNA3 in absence of Ded1 (dark blue) and presence of saturating Ded1 (red). (D) $k_{max}$ observed in absence

*Figure 4 continued on next page*

*Figure 4 continued*

(blue) and presence of saturating Ded1-ΔCTD (purple). (**E**) The average fold change in the $k_{max}(k_{max}^{+Ded1}/k_{max}^{-Ded1})$ observed in presence and absence of WT Ded1 (dark grey) and Ded1-ΔCTD (light grey). Red line indicates rates in absence of Ded1. (**F**) The $K_{1/2}$ of Ded1-ΔCTD value shown as green bars. (**A–F**) Bars indicate mean values calculated from the 3 independent experiments represented by the data points. The superimposed horizontal line (black) indicate the mean maximal rates (**A, D**) or $K_{1/2}$ (**C, F**) observed with WT eIF4E·eIF4G (A, C, from *Figure 3* and *Figure 4—figure supplement 2A*) or WT Ded1 (D, F from *Figures 1* and *2*), and error bars represent 1 SD from the mean (this representation will be referred to as line/whisker plot). See *Figure 4—figure supplements 1* and *2* and *Figure 4—source data 1*.
DOI: https://doi.org/10.7554/eLife.38892.014

The following source data and figure supplements are available for figure 4:

**Source data 1.** (Source data file for *Figure 4*).
DOI: https://doi.org/10.7554/eLife.38892.017

**Figure supplement 1.** Selected effects of the eIF4G-ΔRNA3 and Ded1-ΔCTD truncations on recruitment of native mRNAs.
DOI: https://doi.org/10.7554/eLife.38892.015

**Figure supplement 1—source data 1.** (Source data file for *Figure 4—figure supplement 1*).
DOI: https://doi.org/10.7554/eLife.38892.018

**Figure supplement 2.** (A, B) $K_{1/2}$ values of eIF4E·eIF4G (A) and eIF4A (B) in the absence of Ded1 (dark blue) and presence of saturating Ded1 (red).
DOI: https://doi.org/10.7554/eLife.38892.016

**Figure supplement 2—source data 1.** (Source data file for *Figure 4—figure supplement 2*).
DOI: https://doi.org/10.7554/eLife.38892.019

0.03 min$^{-1}$; WT – 1.3 ± 0.1 min$^{-1}$), compared to the values observed for WT eIF4E·eIF4G (*Figure 4A*, orange bars vs. line/whiskers). Somewhat smaller reductions in $k_{max}$ were observed for *SFT2* and *SFT2-M* (*Figure 4A*). Thus, as summarized in *Figure 4B*, deleting RNA3 nearly eliminated the stimulatory effect of Ded1 on $k_{max}$ values for *RPL41A, HOR7,* and *CP-8.1* (black vs. grey bars vs. dashed red line, the latter indicating no stimulation by Ded1).

Importantly, when Ded1 was replaced with Ded1-ΔCTD in reactions containing WT eIF4E·eIF4G, we observed effects on $k_{max}$ values and rate enhancements very similar to those described above for ΔRNA3 (*Figure 4D–E*), which is consistent with a functional interaction between RNA3 and Ded1-CTD. (We verified that saturating concentrations of eIF4E·eIF4G-ΔRNA3 and Ded1-ΔCTD were used in these experiments by showing that rates of recruitment for *SFT2, RPL41A* or *HOR7* mRNAs were not elevated even at much higher concentrations of the variants (*Figure 4—figure supplement 1B–D*, and data not shown). We also confirmed that Ded1-ΔCTD has RNA-dependent ATPase activity similar to that of the WT Ded1 (*Figure 1—figure supplement 1D–E*)).

To further verify that RNA3 of eIF4G and Ded1-CTD interact in mRNA recruitment, we combined the eIF4E·eIF4G-ΔRNA3 and Ded1-ΔCTD variants in the same assays (at the saturating concentrations determined for each alone). We examined recruitment of *RPL41A*, for which eIF4G-RNA3 and Ded1-CTD were each essential for rate enhancement by Ded1; *CD-8.1*, whose $k_{max}$ was not significantly reduced by eIF4E·eIF4G-ΔRNA3 or Ded1-ΔCTD compared to the corresponding WT proteins; and *CP-8.1*, for which each domain deletion had an intermediate effect on rate enhancement by Ded1. The effects on the $k_{max}$ values for these three mRNAs on combining the eIF4E·eIF4G-ΔRNA3 and Ded1-ΔCTD mutants were similar to what we observed with each deletion individually (*Figure 4—figure supplement 1E*). These data strongly suggest that the changes in $k_{max}$ values conferred by eliminating either eIF4G-RNA3 or Ded1-CTD arise from loss of interaction between these two domains, because once the interaction is disrupted by removing one domain no further defect results from also removing the other.

We next examined whether eliminating eIF4G-RNA3 alters the concentration of eIF4E·eIF4G required to achieve half-maximal rate acceleration, that is, the $K_{1/2}^{eIF4E·eIF4G}$. To this end, we first determined the $K_{1/2}^{eIF4E·eIF4G}$ of WT eIF4E·eIF4G for each mRNA in the presence or absence of Ded1. For four native mRNAs and *SFT2-M*, the presence of Ded1 lowered the $K_{1/2}^{eIF4E·eIF4G}$ by factors of ~2 to 2.5 (*Figure 4—figure supplement 2A*, cols. 1 – 5, red vs. blue), consistent with the idea that Ded1 interacts productively with eIF4E·eIF4G on all five of these mRNAs. Contrary to the natural mRNAs, addition of Ded1 increased the $K_{1/2}^{eIF4E·eIF4G}$ of *-SL* and *CP-8.1* mRNAs, reaching values similar to those observed for the natural mRNAs in the presence of Ded1 (*Figure 4—figure supplement 2A*, cols. 6 and 8, red vs. blue). (The $K_{1/2}^{eIF4E·eIF4G}$ for *CD-8.1* without Ded1 could not be determined

accurately because of its endpoint defects at lower eIF4E·eIF4G concentrations.) Thus, the maximum stimulation of recruitment of the synthetic mRNAs in the absence of Ded1 can be achieved at relatively low eIF4E·eIF4G concentrations, but higher eIF4E·eIF4G concentrations are required to support the additional stimulation of recruitment conferred by Ded1. (See *Figure 4—figure supplement 2* legend for additional comments.)

We then proceeded to determine the effect of eliminating RNA3 on the concentration of eIF4E-·eIF4G required for the maximum recruitment rate in the absence of Ded1. For mRNAs *RPL41A, HOR7, SFT2, SFT2-M,* and *-SL*, the $K_{1/2}$ values for eIF4E·eIF4G-ΔRNA3 did not differ substantially from those of WT eIF4E·eIF4G, although it was ~2 fold higher for *CP-8.1* (*Figure 4C*, blue bars vs. line/whiskers summarizing results for WT eIF4E·eIF4G taken from *Figure 4—figure supplement 2A*). (The $K_{1/2}^{\text{eIF4E·eIF4G}}$ could not be accurately measured for *CD-8.1* using eIF4E·eIF4G-ΔRNA3 due to endpoint defects.) In reactions containing Ded1, by contrast, the $K_{1/2}^{\text{eIF4E·eIF4G}}$ values for eIF4E·eIF4G -ΔRNA3 were increased by 2- to 5-fold relative to the values determined for WT eIF4E·eIF4G for all mRNAs tested (*Figure 4C*, red bars versus superimposed line/whiskers results for WT eIF4E·eIF4G taken from *Figure 4—figure supplement 2A*). Thus, on removal of RNA3, relatively higher concentrations of eIF4E·eIF4G are required to achieve maximal rate stimulation by Ded1, supporting a functionally important interaction between RNA3 and Ded1. We also determined the effects of eliminating the Ded1 CTD on $K_{1/2}$ values for Ded1. Similar to the results obtained for eIF4E·eIF4G-ΔRNA3, higher $K_{1/2}$ values were observed for Ded1-ΔCTD versus WT Ded1 for four of the five mRNAs that exhibit appreciable rate stimulation by Ded1-ΔCTD (which excludes *RPL41A* and *HOR7*) (*Figure 4F*, green bars versus superimposed line/whiskers results for WT Ded1 from *Figures 1C* and *2D*).

In summary, our results indicate that there is an interaction between eIF4G-RNA3 and Ded1-CTD that facilitates Ded1 function in mRNA recruitment to the PIC. The increased $K_{1/2}$ values evoked by eliminating either domain suggests that their interaction enhances assembly of the eIF4G·eIF4E·eIF4A·Ded1 tetrameric complex (*Gao et al., 2016*). The finding that increased concentrations of eIF4E·eIF4G-ΔRNA3 or Ded1-ΔCTD can rescue the rate ($k_{max}$) defects caused by the domain deletions for some mRNAs but not for others suggests that in certain mRNA contexts this interaction plays a role in addition to simply promoting interaction between eIF4G and Ded1.

## The RNA2 domain of eIF4G functions in Ded1-dependent mRNA recruitment

Ded1 also interacts with the RNA2 domain of eIF4G (*Senissar et al., 2014*), but the importance of this interaction for Ded1 function is unknown. Comparing the eIF4E·eIF4G-ΔRNA2 variant (with an internal deletion of RNA2) to WT eIF4E·eIF4G, we found that, as for RNA3, deletion of RNA2 influenced recruitment of the mRNAs to different extents. In reactions lacking Ded1, we observed no significant differences in $k_{max}$ for any of the seven mRNAs examined (*Figure 5A*, blue bars vs. line/whiskers for WT eIF4E·eIF4G data from *Figure 3*, blue). By contrast, removal of RNA2 increased the $K_{1/2}$ for eIF4E·eIF4G-ΔRNA2 versus WT eIF4E·eIF4G by 3 – 5-fold for *SFT2-M, -SL,* and *CP-8.1* mRNAs in reactions lacking Ded1 (*Figure 5C*, blue bars vs. line/whiskers from *Figure 4—figure supplement 2A*). ΔRNA2 also conferred an endpoint defect for *SFT2* at lower concentrations, precluding determination of its effects on the $K_{1/2}$ for eIF4E·eIF4G. (Because of endpoint defects for *CD-8.1* even with WT eIF4E·eIF4G, the importance of RNA2 cannot be evaluated for this mRNA.) Considering that ΔRNA3 increased the $K_{1/2}$ for eIF4E·eIF4G only for *CP-8.1* in reactions lacking Ded1, it appears that RNA2 has relatively more important Ded1-independent functions than RNA3 in recruitment of particular mRNAs.

Stronger effects of ΔRNA2 were observed in reactions containing Ded1, markedly reducing the $k_{max}$ for the two mRNAs containing cap-distal SLs, *SFT2* ($k_{max}$ of 1.4 ± 0.1 min$^{-1}$ vs. 3.4 ± 0.1 min$^{-1}$ for WT) and *CD-8.1* ($k_{max}$ of 0.7 ± 0.03 min$^{-1}$ vs. 1.9 ± 0.2 min$^{-1}$ for WT), with smaller reductions for *SFT2-M* and *CP-8.1* (*Figure 5A*, orange bars vs. line/whiskers; and *Figure 5B*, black vs. grey bars vs. dotted red line). By contrast, RNA2 was dispensable for maximal Ded1 acceleration of *RPL41A, HOR7,* and *-SL* mRNA recruitment (*Figure 5B*, black vs. grey bars). It is noteworthy that *RPL41A* and *HOR7* exhibited the strongest dependence on RNA3 (*Figure 4B*), but were insensitive to loss of RNA2 (*Figure 5B*) for maximal rate stimulation by Ded1.

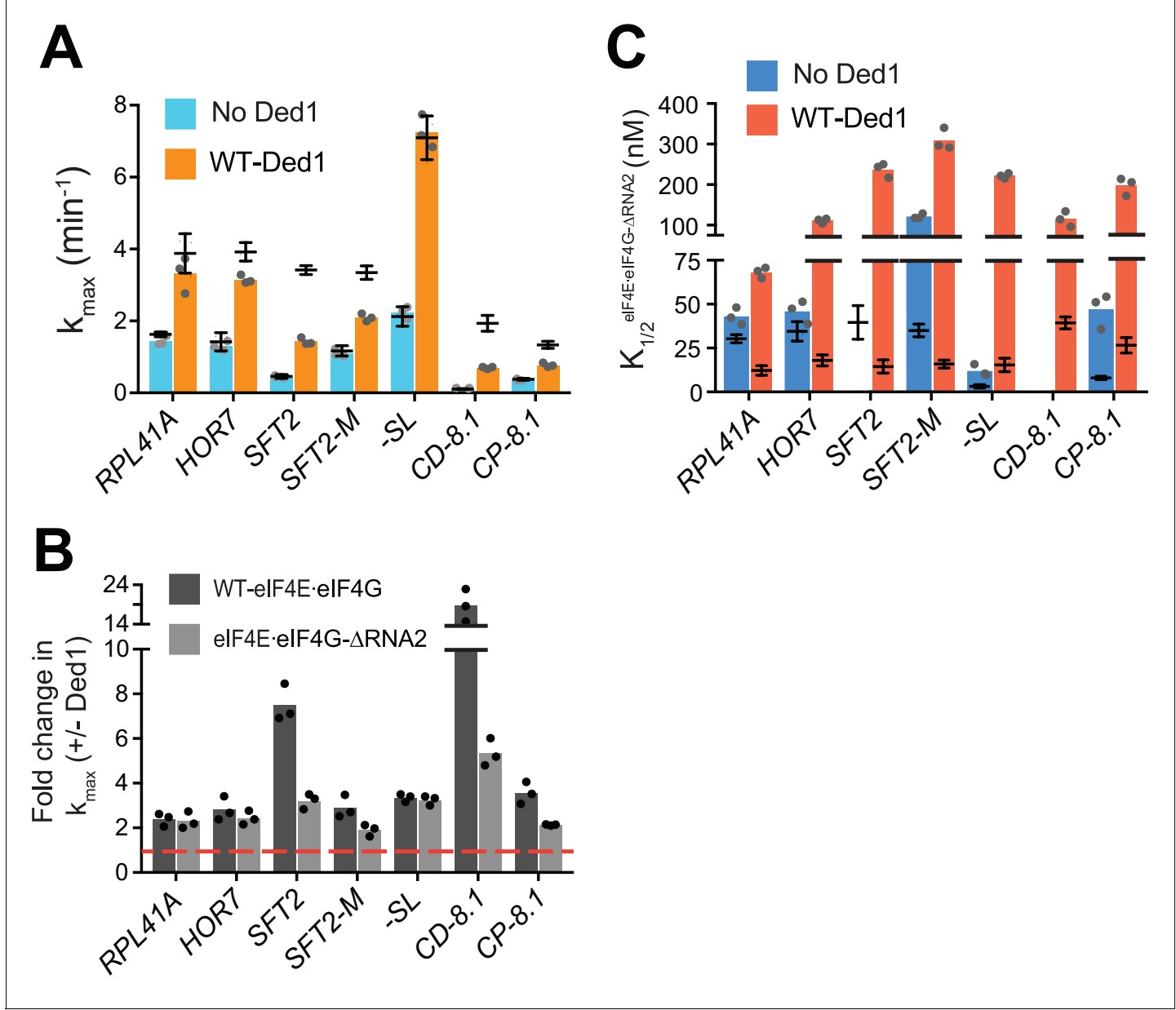

**Figure 5.** RNA2 domain of eIF4G is crucial for Ded1-dependent stimulation of mRNA recruitment. (A) $k_{max}$ in the absence (blue) and presence of saturating Ded1 (orange) with eIF4E·eIF4G-ΔRNA2. (B) The average fold-change in the maximal rate observed in the presence and absence of Ded1 with WT eIF4E·eIF4G (dark grey) and eIF4E·eIF4G-ΔRNA2 (light grey). Red line indicates no change in the rate on Ded1 addition. (C) The $K_{1/2}$ of eIF4E·eIF4G-ΔRNA2 in the absence of Ded1 (dark blue) and presence of saturating Ded1 (red). (A–C) Bars represent mean values (n = 3) and points on each bar show the individual experimental values. The line and whisker plot indicate the mean maximal rates (A) or $K_{1/2}$ (C) observed with WT eIF4E·eIF4G, and error bars represent 1 SD, as depicted in **Figure 4**. See **Figure 4—figure supplements 1** and **2** and **Figure 5—source data 1**.
DOI: https://doi.org/10.7554/eLife.38892.020

The following source data is available for figure 5:

**Source data 1.** (Source data file for **Figure 5**).
DOI: https://doi.org/10.7554/eLife.38892.021

All of the mRNAs exhibited increases in $K_{1/2}$ for eIF4E·eIF4G-ΔRNA2 versus WT eIF4E·eIF4G (3 – 20-fold; **Figure 5C**, red bars vs. line/whiskers). Thus, eliminating RNA2 increases the concentration of eIF4E·eIF4G required for maximal Ded1 stimulation for all mRNAs tested, which might reflect its importance in promoting formation of the eIF4G·eIF4E·eIF4A·Ded1 complex, in the manner

concluded above for RNA3. Elevated concentrations of eIF4E·eIF4G-ΔRNA2 enable maximum recruitment rates similar to those achieved with WT eIF4E·eIF4G for the mRNAs with lower degrees of structure – *RPL41A, HOR7,* and *-SL*; whereas the more structured mRNAs display varying reductions in $k_{max}$ at saturating eIF4E·eIF4G-ΔRNA2 concentrations, with the two mRNAs harboring cap-distal stem loops – *SFT2* and *CD-8.1* – having the largest rate enhancement defects. This suggests that RNA2 enhances Ded1 function on the structured mRNAs beyond its ability to simply stabilize Ded1-eIF4G interaction.

## The N-terminal domain of Ded1 enhances mRNA recruitment

It was shown previously that the N-terminal domain (NTD) of Ded1 physically interacts with eIF4A, and is required for eIF4A stimulation of Ded1 unwinding activity in vitro (*Gao et al., 2016*; *Senissar et al., 2014*). We tested the effects of Ded1 on $K_{1/2}$ of eIF4A and, as observed with WT eIF4E·eIF4G, Ded1 influenced $K_{1/2}^{eIF4A}$ differently on these mRNAs, providing evidence that Ded1 has functional interactions with eIF4A during mRNA recruitment (*Figure 4—figure supplement 2B*). Hence, we examined the effect of eliminating the Ded1-NTD on recruitment of our panel of mRNAs. The $k_{cat}$ and $K_m$ values for RNA-dependent ATP hydrolysis were indistinguishable between WT Ded1 and the ΔNTD variant (*Figure 1—figure supplement 1D–E*). For the *SFT2, CD-8.1,* and *CP-8.1* mRNAs, which harbor defined SLs in their 5'-UTRs, eliminating the Ded1 NTD decreased $k_{max}$ by 1.5 – 2-fold, whereas the $k_{max}$ values for *RPL41A, HOR7, SFT2-M,* and *-SL* mRNAs were not significantly altered by ΔNTD (*Figure 6A*, purple bars vs. line/whiskers for WT Ded1; and *Figure 6B*). However, 1 – 2 orders of magnitude higher $K_{1/2}$ values for the Ded1-ΔNTD versus WT Ded1 were observed with all mRNAs except *CD-8.1* (*Figure 6C*, green bars vs. line/whiskers for WT Ded1). Thus, removing the Ded1 NTD significantly increases the concentrations of Ded1 required to achieve enhancement of recruitment of six out of the seven mRNAs tested. These data are consistent with the idea that interaction of eIF4A and the Ded1 NTD enhances assembly or stability of the eIF4G·eIF4E·eIF4A·Ded1 complex, and stimulates Ded1 helicase activity; although the Ded1-NTD might also mediate important interactions with other components of the system. As with deletions of the eIF4G RNA domains and Ded1-CTD, mRNA-specific defects were conferred by deleting the Ded1 NTD.

## Discussion

Employing a purified yeast translation initiation system, we reconstituted the function of DEAD-box helicase Ded1 in stimulating the rate of 48S PIC assembly on both native and model mRNAs. This stimulation in vitro recapitulates the Ded1-dependence of translation of mRNAs observed in vivo using ribosome profiling, in which mRNAs having longer and more structured 5'-UTRs display hyper-dependence on Ded1 relative to mRNAs with shorter and less structured 5'-UTRs (*Sen et al., 2015*). We showed that defined SL structures both decrease rates of 48S PIC assembly in the absence of Ded1 and increase the fold-stimulation afforded by Ded1. These results provide direct biochemical evidence supporting the proposition that Ded1 enhances translation initiation in vivo by resolving secondary structures formed by 5'-UTR sequences. Our results also showed that Ded1-accelerated recruitment of several mRNAs depends completely on the presence of eIF4E·eIF4G, and that domains mediating Ded1 interactions with eIF4G or eIF4A enhance Ded1 stimulation of 48S PIC assembly for all mRNAs tested, consistent with the previous work indicating that Ded1 binding to the eIF4G·eIF4E·eIF4A complex enhances its activity in RNA unwinding assays (*Gao et al., 2016*). However, Ded1 can also stimulate recruitment of some mRNAs in the absence of eIF4E·eIF4G, indicating that it can act independently of eIF4F as well.

### DEAD-box proteins Ded1 and eIF4A have complementary but distinct functions in mRNA recruitment

Inactivation of conditional mutants of either Ded1 or eIF4A in vivo results in strong reduction in bulk polysomes and decreased expression of reporter mRNAs bearing unstructured 5'-UTRs (*Chuang et al., 1997*; *de la Cruz et al., 1997*; *Sen et al., 2015*), indicating that both proteins are important for translation of most mRNAs. These general reductions of translation are masked in ribosome profiling studies because the ribosomal footprint and mRNA read counts must be normalized

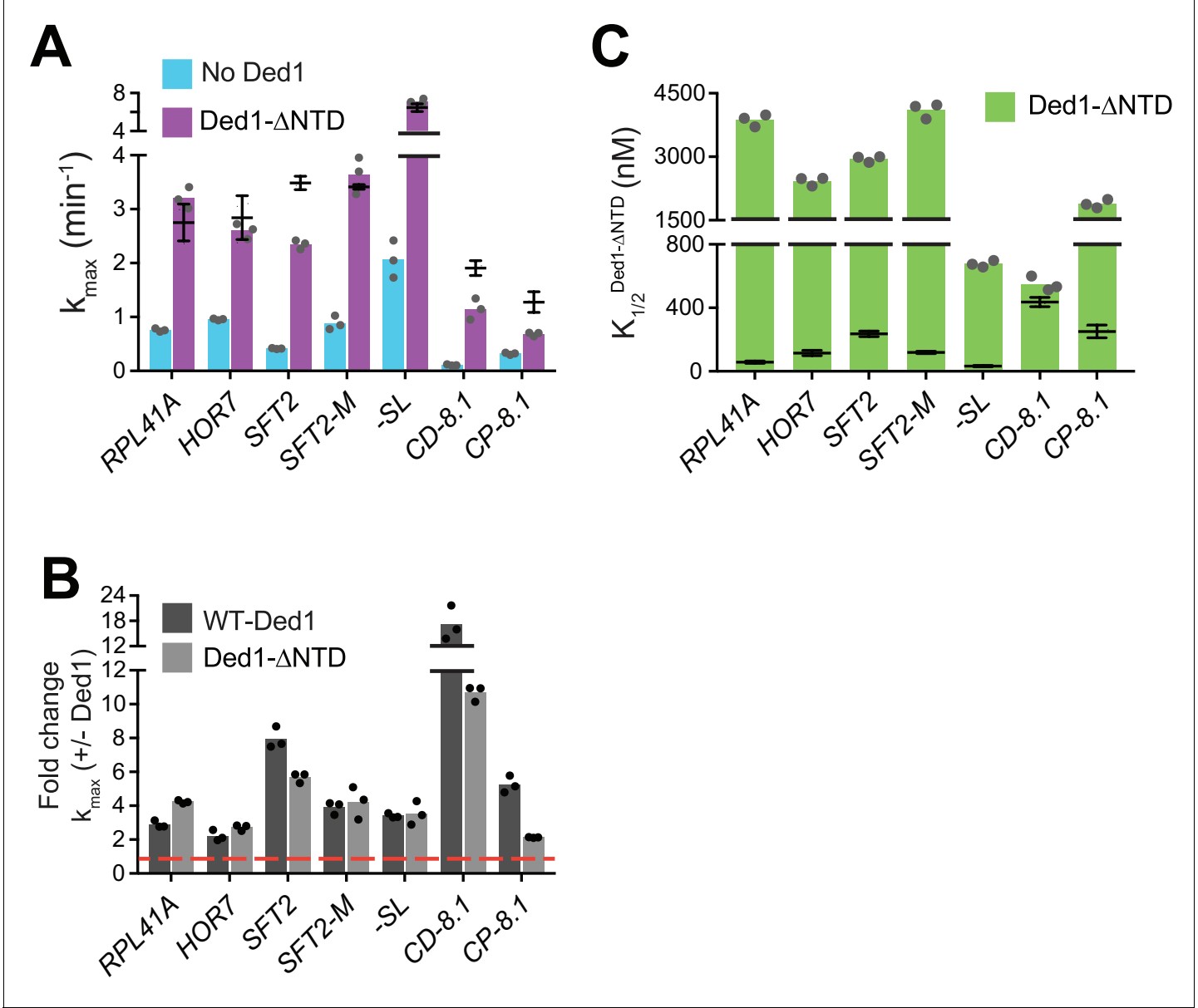

**Figure 6.** Ded1-NTD enhances mRNA recruitment. (**A**) $k_{max}$ values observed in the absence (blue) and presence of saturating Ded1-$\Delta$NTD (purple). (**B**) The average fold change in the maximal rate ($k_{max}^{+Ded1}/k_{max}^{-Ded1}$) observed in the presence and absence of WT Ded1 (dark grey) and Ded1- $\Delta$NTD (light grey). Red line indicates no change in the rate with Ded1. (**C**) $K_{1/2}$ of Ded1-$\Delta$NTD value shown as green bars. (**A–C**) Bars represent the mean values (n = 3) and points on each bar show the individual experimental values. The line and whisker plot indicate the mean maximal rates (**A**) or $K_{1/2}$ (**C**) observed with WT Ded1, and error bars represent 1 SD. See *Figure 4—figure supplements 1* and *2* and *Figure 6—source data 1*.

DOI: https://doi.org/10.7554/eLife.38892.022

The following source data is available for figure 6:

**Source data 1.** (Source data file for *Figure 6*).

DOI: https://doi.org/10.7554/eLife.38892.023

to the total reads obtained in each sample/strain, such that the TE change for each mRNA is determined relative to the average TE change for all mRNAs examined in each sample/strain. mRNAs judged to hyperdependent or hypodependent on Ded1 in ribosome profiling experiments exhibit larger or smaller than average reductions in relative TE, respectively, but they may all exhibit decreased absolute TEs in *ded1* vs. WT cells (*Sen et al., 2015*). The ribosome profiling analysis of

*ded1* mutants revealed that ~10% of all mRNAs – particularly those with long, structured 5'-UTRs – exhibit greater than average TE reductions in the *ded1* vs. WT cells, and were designated as Ded1-hyperdependent. Consistent with a stimulatory role for Ded1 in translation of most mRNAs, we observed here that Ded1 increases the maximal rate of 48S PIC formation by ~2 – 3 fold on native Ded1-hypodependent mRNAs or mRNAs with 5'-UTRs of low structural complexity (*RPL41A*, *HOR7*, and *SFT2-M*), as well as on a synthetic mRNA with an unstructured 5'-UTR (-*SL*) (*Figures 1B* and *2C*). Importantly, Ded1 conferred much greater acceleration of 48S PIC assembly on all four Ded1-hyperdependent mRNAs examined (*Figure 1B*).

eIF4A also enhances the translation of nearly all mRNAs in vivo, although, unlike Ded1 where sizable sets of mRNAs are hyper- or hypo-dependent on its function, most mRNAs are similarly (strongly) dependent on eIF4A for translation (*Firczuk et al., 2013*; *Sen et al., 2015*). In line with these in vivo observations, in the reconstituted system eIF4A promotes 48S PIC assembly on all mRNAs tested, increasing the $k_{max}$ for the synthetic mRNA with unstructured 5'-UTR (-*SL*) by 60-fold and even accelerating recruitment of completely unstructured model mRNAs by $\geq$7-fold (*Yourik et al., 2017*). However, although Ded1 and eIF4A both facilitate recruitment of most mRNAs, and both are essential in vivo (*Chuang et al., 1997*; *Linder and Slonimski, 1989*); their functions are distinct. Ded1 cannot substitute for eIF4A in vitro (*Figure 3—figure supplement 1*), but it promoted recruitment of all mRNAs tested beyond the level achieved by saturating concentrations of eIF4A and eIF4E·eIF4G (*Figures 1B* and *2C*).

We previously proposed that eIF4A stimulates a step of 48S PIC assembly common to all mRNAs, such as disrupting the ensemble of weak RNA-RNA interactions that impede PIC attachment to the 5'-UTR or subsequent scanning (*Yourik et al., 2017*). In addition, eIF4A might also directly promote loading of mRNA onto the PIC, for example by modulating conformational changes in the 40S subunit or by threading the 5'-end into the mRNA binding channel (*Kumar et al., 2016*; *Sokabe and Fraser, 2017*). In common with eIF4A, Ded1 may enhance recruitment of all mRNAs by disrupting their global structures created by dynamic ensembles of base-pairing throughout their lengths. Unlike eIF4A, however, Ded1 can efficiently resolve more stable structures, including local stem-loops—achieving an order-of-magnitude acceleration for mRNAs with the most structured 5'-UTRs. If the proposed Ded1 function in promoting 48S PIC formation by disrupting global mRNA structure requires lower Ded1 concentrations than its role in resolving more stable structures within or involving the 5'-UTR, it would be consistent with our findings that mRNAs with SLs require higher Ded1 concentrations to achieve the much greater fold-stimulation of 48S assembly afforded by Ded1 compared to mRNAs lacking SLs (*Figures 1* and *2*).

## Evidence supporting the functional importance of an eIF4G·eIF4E·eIF4A·Ded1 tetrameric complex

Ded1 alters the $K_{1/2}$ of eIF4E·eIF4G and eIF4A for most mRNAs (*Figure 4—figure supplement 2*), and these new $K_{1/2}$ values may signify the changes in the concentrations of eIF4A and eIF4E·eIF4G required for proper assembly of the eIF4G·eIF4E·eIF4A·Ded1 tetrameric complex on each mRNA. The deleterious effects of eliminating known interactions between Ded1 and eIF4G or eIF4A further suggests the importance of the tetrameric complex formation for robust Ded1 function. With only two exceptions (*CP-8.1* for Ded1-ΔCTD and *CD-8.1* for the Ded1-ΔNTD), we found that deleting the RNA2 or RNA3 domain of eIF4G, or the CTD or NTD of Ded1, increased the concentrations of the corresponding eIF4G or Ded1 variants required to achieve the half-maximal rate of 48S PIC assembly (ie., their $K_{1/2}$ values) on each mRNA examined, as summarized by the heatmap in *Figure 7A*. This holds for the mRNAs with the shortest or least structured 5'-UTRs (-*SL*, *RPL41A*, and *HOR7*) as well as those with the most highly structured 5'-UTRs (*SFT2*, *CD-8.1*, and *CP-8.1*). Because all of these domain deletions abrogate known interactions linking Ded1 to eIF4G or eIF4A, a plausible way to account for these findings is to propose that, regardless of the amount of secondary structure in the mRNA 5'-UTR, rapid mRNA recruitment depends on Ded1 functioning within the eIF4E·eIF4G·eIF4A·Ded1 complex; and that eliminating any interaction between Ded1 and eIF4G or eIF4A necessitates a higher concentration of the mutant variant for efficient complex formation. Judging by the magnitude of the increases in $K_{1/2}$ conferred by eliminating different domains (*Figure 7A*), it would appear that eIF4G-RNA2 and Ded1-NTD are generally more important than the eIF4G-RNA3/Ded1-CTD duo in promoting assembly or stability of the eIF4E·eIF4G·eIF4A·Ded1 complex. However, we cannot rule out the possibility that the domain deletions also impair a

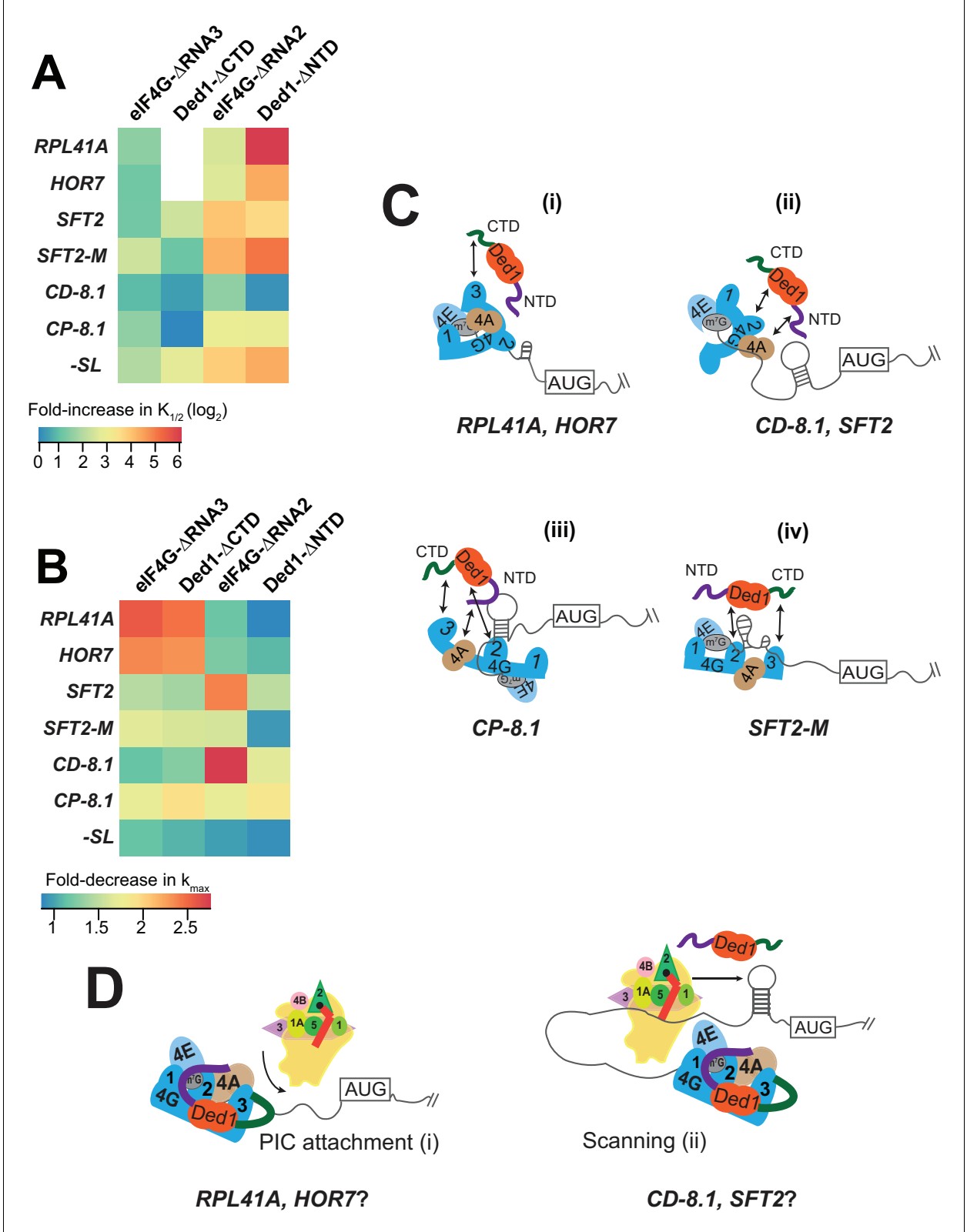

**Figure 7.** Models for mRNA-specific eIF4F-Ded1 interactions. (**A**) Heatmap representation of $\log_2$ fold-increases in $K_{1/2}$ values of eIF4E·eIF4G-ΔRNA3 or eIF4E·eIF4G-ΔRNA2 versus WT eIF4E·eIF4G; and of Ded1-ΔCTD or Ded1-ΔNTD versus WT Ded1, calculated from data in *Figure 4C and 5C* (red bars vs. line/whiskers) or *Figure 4F and 6C* (green bars vs. line/whiskers). (**B**) Heatmap depicting the fold-changes in $k_{max}$ values of the indicated eIF4E·eIF4G or Ded1 truncations($k_{max}^{mutant}$ vs $k_{max}^{WT}$) calculated from *Figures 4A, D, 5A* and *6A* (orange or purple bars vs line/whiskers). (**C**) 'mRNA

*Figure 7 continued on next page*

Figure 7 continued

geometry' model depicting how different mRNAs exhibiting distinct configurations of the occurrence and location of RNA structures (shown as hairpins or stem-loops) could influence the relative importance of different domain interactions linking Ded1 to eIF4G or eIF4A within the eIF4G·eIF4E·eIF4A·Ded1 tetrameric complex. (D) 'Kinetic' model depicting how different mRNAs might differ in the extent to which PIC attachment or scanning are the rate-limiting steps in 48S PIC assembly. Depending on which step is rate-limiting, the requirements for Ded1, either acting alone or within the eIF4G·eIF4E·eIF4A·Ded1 complex, could be different on different mRNAs.

DOI: https://doi.org/10.7554/eLife.38892.024

different interaction, for instance, with mRNA or another factor such as eIF4B or eIF3 that is crucial for rapid mRNA recruitment (*Mitchell et al., 2010*).

## Ded1 and eIF4G mutants affect the maximal rates for recruitment differently depending on the mRNA

With some mRNAs, the maximum rate of recruitment observed with WT Ded1 and WT eIF4E·eIF4G ($k_{max}$) could also be achieved using elevated concentrations of the mutant variant; whereas in other cases, the observed $k_{max}$ was diminished from the WT value even at saturating amounts of the mutant Ded1 or eIF4G variant. We regard such reductions in $k_{max}$ as indicating impairment of a fundamental role of the deleted eIF4G or Ded1 domain in rapid recruitment of the affected mRNA. Accordingly, we summarized these effects for each factor truncation in a heatmap (*Figure 7B*) to evaluate the requirements for particular domains or interactions for maximum rate stimulation by Ded1 with each mRNA tested.

It is evident from the heatmap that deletion of the eIF4G-RNA3 or Ded1-CTD domain confers a wide range of $k_{max}$ reductions: 2 – 3-fold for *RPL41A* and *HOR7*; ~1.5 fold for *SFT2, SFT2-M,* and *CP-8.1*, and almost no change for *-SL* and *CD-8.1* (*Figure 7B*, cf. cols. 1 – 2, all rows). Importantly, however, in all cases the effects of deleting eIF4G-RNA3 and Ded1-CTD are similar for a given mRNA, supporting the proposition that these effects result from loss of the eIF4G-RNA3/Ded1-CTD interaction. Eliminating this interaction essentially abolishes the rate enhancement provided by Ded1 for recruitment of *RPL41A* and *HOR7* mRNAs (*Figure 4B,E* and *Figure 4—figure supplement 1E*), indicating that the Ded1-CTD/eIF4G-RNA3 interaction is essential for the eIF4E·eIF4G·eIF4A·Ded1 complex (i.e., at saturating Ded1 concentration) to accelerate a slow step in 48S PIC assembly on these mRNAs (*Figure 7C(i)*). This step is apparently enhanced through different interactions or is less rate-limiting for the other mRNAs tested.

We observed a similar diversity of effects on $k_{max}$ values depending on the mRNA for the eIF4G-RNA2 and Ded1-NTD deletions. ΔRNA2 had little effect on $k_{max}$ for *RPL41A* and *HOR7*, in contrast to the deleterious effects of ΔRNA3 for these two mRNAs. Similarly, ΔRNA2 markedly reduced the $k_{max}$ for *SFT2* and *CD-8.1* by ~2.5 – 3-fold, whereas ΔRNA3 had little effect on these mRNAs (*Figure 7B*, col. 3 vs. col. 1). Similar to ΔRNA3 however, ΔRNA2 decreased the $k_{max}$ values for *SFT2-M* and *CP-8.1* by ~1.5 – 2-fold, with minimal effect on *-SL* mRNA (*Figure 7B*). In this case, the RNA2 domain appears to be most important for the ability of the eIF4E·eIF4G·eIF4A·Ded1 complex to stimulate recruitment of the two mRNAs with cap-distal SLs, *SFT2* and *CD-8.1*; nearly dispensable for the mRNAs with the lowest degrees of structure, *-SL, RPL41A* and *HOR7*; and of intermediate importance for *CP-8.1* and *SFT2-M*. These observations suggest that RNA2 facilitates Ded1 function in melting out structures encountered by the PIC during attachment or scanning (*Figure 7C(ii-iv)*). (Although *SFT2-M* lacks the major cap-distal SL in WT *SFT2*, it contains an additional cap-proximal structure in vitro that might underlie its greater dependence on RNA2 versus *-SL, RPL41A,* and *HOR7*.)

Whereas deletion of the Ded1-NTD had the largest effect of the four eIF4G or Ded1 truncations on $K_{1/2}$ values (*Figure 7A*), it had the smallest effects on maximal rates of recruitment, reducing $k_{max}$ values between ~1.5 to~2 fold for *SFT2, CD-8.1,* and *CP-8.1*, but having little effect on the other mRNAs (*Figure 7B*). As all three affected mRNAs have stable SLs and the unaffected mRNAs do not, these data might indicate that the Ded1-NTD, presumably by interacting with eIF4A, modestly enhances the ability of the eIF4E·eIF4G·eIF4A·Ded1 complex to unwind stable secondary structures—complementing the function of eIF4G-RNA2 in this reaction (*Figure 7C(ii-iii)*).

What is perhaps most striking about the effects of the eIF4G and Ded1 truncations on both the $K_{1/2}$ (*Figure 7A*) and $k_{max}$ (*Figure 7B*) values is that mRNAs have distinct patterns of responses. The

various eIF4G-Ded1 domain interactions affect *CP-8.1* and *CD-8.1* recruitment quite differently even though these mRNAs differ only by the location of the same SL in an unstructured 5'-UTR. This suggests that there is not a single, uniform mechanism through which Ded1 operates on all mRNAs; instead the diversity of structures in mRNAs requires that Ded1 and the eIF4E·eIF4G·eIF4A·Ded1 complex can operate in multiple modes. The structural diversity inherent in mRNAs presents a challenge for the translational machinery because, once the eIF4F complex attaches to the 5'-cap, structural elements could be oriented in a variety of locations in three-dimensional space relative to its functional domains (*Figure 7C*). This problem could explain why eIF4G is so large and flexible and has multiple RNA- and factor-binding domains, which might confer sufficient plasticity to interact with mRNA structures presented in a variety of orientations and distances. Likewise, the multiple Ded1 binding domains on eIF4G might allow Ded1 to assume different positions relative to the diverse mRNA structures it encounters on different mRNAs. The mRNA specificity of effects of the truncation mutants of Ded1 and eIF4G on both $K_{1/2}$ and $k_{max}$ values are consistent with the notion that the eIF4E·eIF4G·eIF4A·Ded1 complex can interact with and modulate the structures of mRNAs in different ways, with the mRNA structure dictating the particular interactions of Ded1 with eIF4G or eIF4A that are most crucial for rapid recruitment.

It is likely that the rate-limiting step(s) for 48S PIC formation will also vary depending on the unique structural features of the mRNA. For some mRNAs, PIC attachment to the 5'-UTR might be rate-limiting because of structures proximal to the 5'-end or because the 5'-end is occluded within the global structure of the mRNA (*Figure 7D* (i)). For other mRNAs, PIC attachment might be relatively fast, but scanning to the start codon could be impeded by stable structures that require Ded1 in the context of the eIF4E·eIF4G·eIF4A·Ded1 complex to resolve (*Figure 7D* (ii)). Since the *RPL41A* and *HOR7* mRNAs have short and less-structured 5'-UTRs, it is plausible that PIC attachment could be rate-limiting on these two mRNAs (*Figure 7D* (i)), whereas scanning of the 5'-UTR could be rate-limiting on the two cap-distal SL-containing mRNAs, *SFT2* and *CD-8.1* (*Figure 7D* (ii)).

Recruitment of some mRNAs was accelerated by Ded1 in the absence of eIF4E·eIF4G, including *SFT2*, *SFT2-M*, *OST3*, and *CD-8.1* (*Figure 3C–E,G*) indicating that Ded1 is also capable of stimulating one or more aspects of 48S PIC assembly outside of the context of the eIF4E·eIF4G·eIF4A·Ded1 complex on certain mRNAs. The three mRNAs that most clearly exhibit complete dependence on eIF4E·eIF4G for Ded1 stimulation, *-SL*, *RPL41A*, and *HOR7*, have low degrees of structure in their 5'-UTRs. Hence, an intriguing possibility is that for mRNAs lacking strong local secondary structure in the 5'-UTR, Ded1 is only needed to promote eIF4F binding to the cap or initial attachment of the PIC at the 5'-end, and this process requires direct interaction of Ded1 with eIF4F at the mRNA 5'-end (*Figure 7D* (i)). For other mRNAs harboring strong local structures in the 5'-UTR, in addition to acting in a eIF4F-Ded1 tetrameric complex to facilitate PIC attachment, Ded1 might unwind these structures and promote scanning independently of its association with eIF4F (*Figure 7D* (ii)). The apparent inability of Ded1 to accelerate recruitment of *CP-8.1* independently of eIF4E·eIF4G might be explained by noting that the SL in this mRNA is cap-proximal, which could require eIF4F-Ded1 interaction for unwinding; however, because no recruitment of this mRNA was observed in the absence of eIF4E·eIF4G or Ded1, it is possible that Ded1 can actually accelerate *CP-8.1* recruitment on its own but the rate is too low to be detected in the absence of eIF4E·eIF4G. Additionally, Ded1 can interact with other factors such as eIF4A, the 40S ribosomal subunit, or the mRNA itself, which might aid in the recruitment of these mRNAs without eIF4E·eIF4G (*Gao et al., 2016*; *Guenther et al., 2018*).

The two different models ('mRNA geometry' and 'rate-limiting steps', *Figure 7C–D*) we are considering to explain the differential effects of the eIF4G and Ded1 domain deletions on different mRNAs are not mutually exclusive. In fact, the proposal that the domains have some specificity for mediating PIC attachment versus scanning probably requires that they localize Ded1 to different parts of the mRNA because the former reaction would occur closer to the 5'-end whereas the latter would occur distal to it. The length and flexibility of eIF4G, coupled with the complex network of interactions possible among eIF4G, eIF4E, eIF4A, Ded1 and mRNA, could have evolved to support the plasticity required to deal with the wide variety of mRNA shapes, sizes and structures that must be loaded onto PICs for translation in eukaryotic cells where transcription and translation are uncoupled.

mRNAs can form long-range interactions between their 5'-UTRs and coding sequences or 3'-UTRs, but because our reporter mRNAs consisted of only 5'-UTRs and the first 60 nucleotides of

their coding sequences, interactions of this kind would not be recapitulated in our system. Therefore, the role of Ded1 in resolving such long-range interactions remains to be elucidated. Additionally, our reporter mRNAs lack 3'-poly(A) tails and the recruitment assay was performed in the absence of the poly(A) binding protein (PABP). PABP can interact with eIF4G, and it would thus be useful in the future to explore how PABP-eIF4G interactions influence Ded1-eIF4G interactions and Ded1 functions in mRNA recruitment and scanning.

## Materials and methods

### Preparation of mRNAs and charged initiator tRNA

Plasmids for in vitro run-off mRNA transcription of Ded1-hypodependent and -hyperdependent mRNAs were constructed using Gibson assembly (*Gibson et al., 2009*). The 5'-UTR and first 60 nucleotides of the coding region of *OST3, SFT2, PMA1, HOR7*, and *FET3* genes were PCR amplified from yeast genomic DNA (BY4741) and cloned into pBluescript II KS + vector (Stratagene) using NEBuilder HiFi assembly according to the manufacturer's instructions (New England Biolabs). In all mRNAs constructs, Xma1 restriction site was added at the end of the coding region during cloning to linearize the plasmids, and two G nucleotides were added at the beginning of the 5'-UTR to improve transcription efficiency. Plasmids for transcription of *SFT2-M, SFT2-M2*, and *PMA1-M* mRNAs were derived from *SFT2* and *PMA1* plasmids, respectively, by mutating their 5'-UTRs (Genscript Corp.). Plasmids for transcription of *RPL41A*, synthetic mRNAs with 5'-UTR consisting of CAA repeats and *RPL41A ORF* and 3'-UTR, and initiator tRNA were described previously (*Acker et al., 2007*; *Mitchell et al., 2010*; *Yourik et al., 2017*). mRNAs and initiator tRNA were transcribed by run-off transcription using T7 RNA polymerase and gel purified as described previously (*Acker et al., 2007*; *Mitchell et al., 2010*). mRNAs were capped ($m^7$GpppG) using either $\alpha-^{32}$P radiolabeled GTP (Perkin Elmer) or unlabeled GTP and vaccinia virus capping enzyme (*Mitchell et al., 2010*). Initiator tRNA was methionylated in vitro using methionine and *E. coli* methionyl-tRNA synthetase, and charged Met-tRNA$_i^{Met}$ was purified of contaminating nucleotides over a desalting column (*Walker and Fredrick, 2008*; *Yourik et al., 2017*).

### Purification of translation initiation factors

Eukaryotic initiation factors- eIF1, eIF1A, eIF2, eIF3, eIF4A, eIF4B, eIF4G·4E (WT and mutants), eIF5- were expressed and purified as described previously (*Acker et al., 2007*; *Mitchell et al., 2010*; *Rajagopal et al., 2012*). 40S ribosomal subunits were prepared as described in (*Munoz et al., 2017*). Recombinant Ded1 proteins (N-terminal His$_6$-tag, pET22b vector)- WT Ded1(1-604), Ded1$^{E307A}$, Ded1-ΔCTD (1-535) and Ded1-ΔNTD (117-604) were purified as described previously (*Gao et al., 2016*; *Hilliker et al., 2011*) with some modifications. Ded1 proteins were expressed in E. coli BL21(DE3) RIL CodonPlus cells (Agilent). Cells were grown at 37°C till OD600 of 0.5, cooled to 22°C, and induced with 0.5 mM IPTG overnight. Cells were re-suspended in the lysis buffer (10 mM HEPES-KOH, pH-7.4, 200 mM KCl, 0.1% IGEPAL CA-630, 10 mM imidazole, 10% glycerol, 10 mM 2-mercaptoethanol, DNaseI (1 U/ml) and cOmplete protease inhibitor cocktail (Roche)), and lysed using a French Press. Ded1 was purified over a nickel column (5 ml His-Trap column, GE Healthcare) followed by phosphocellulose chromatography (P11, Whartman). Purified protein was dialyzed into dialysis buffer (10 mM HEPES-KOH, pH 7.4, 200 mM KOAc, 50% Glycerol, 2 mM DTT) and stored at −80°C. Ded1-ΔNTD was purified with the same method as the wild-type Ded1 with the following modifications. The N-terminal His$_6$-SUMO tag was removed by incubating nickel-column purified protein with a His$_6$-SUMO protease (McLab) at 4°C overnight, followed by second round of nickel column purification (*Gao et al., 2016*).

### mRNA recruitment assay

48S PICs were assembled and native gel shift assay was performed as described previously (*Mitchell et al., 2010*; *Yourik et al., 2017*) with following modifications. Reactions were assembled in 1X Recon buffer (30 mM HEPES-KOH, pH 7.4, 100 mM KOAc, 3 mM Mg(OAc)$_2$, and 2 mM DTT) containing 300 nM eIF2, 0.5 mM GDPNP·Mg$^{2+}$, 200 nM Met-tRNAi$^{Met}$, 1 µM eIF1, 1 µM eIF1A, 300 nM eIF5, 300 nM eIF4B, 300 nM eIF3, 30 nM 40S subunits, eIF4A, eIF4E·eIFG, and Ded1. The non-hydrolyzable GTP analog GDPNP was used in forming the TC to stabilize the 43S and 48S

complexes by preventing conversion to the eIF2·GDP state. The concentrations of eIF4A, eIF4E··eIFG, and Ded1 varied for recruitment of different mRNAs. eIF4A: *RPL41A, HOR7, -SL, CP-8.1*, and *CD-8.1* = 7 µM; *SFT2, SFT2-M, SFT2-M2, OST3, PMA-1, PMA-1M* and *FET3* = 15 µM. eIF4E·eIFG = 75 nM for all mRNAs, except *OST3* (150 nM). Ded1: *RPL41A* and *-SL* = 250 nM, *HOR7* = 500 nM, all other mRNAs = 1 µM. The concentrations of initiation factors were saturating except for the titrant. Reactions were incubated at 26?C for 10 min, and were initiated by simultaneous addition of 15 nM $^{32}$P-m$^7$G mRNA and 5 mM ATP·Mg$^{2+}$. For kinetic analysis, 2 µl aliquot were removed at appropriate times, reactions were stopped by addition of 600–1000 nM non-radiolabeled m$^7$G-mRNA (same mRNA as the $^{32}$P-m$^7$G mRNA), and loaded onto a 4% non-denaturing PAGE gel to separate 48S PICs from the free mRNA. Percentage of mRNA recruited to the 48S PIC was calculated using ImageQuant software (GE Healthcare). Data were fitted with a single exponential rate equation to calculate apparent rate of recruitment using KaleidaGraph software (Synergy). Apparent rates were plotted against the concentration of the titrant and fitted with hyperbolic equation to calculate the maximal rates of recruitment and the concentration of the titrant required to achieve the half-maximal rates ($K_{1/2}$). To measure the maximal endpoints of recruitment, the reactions were incubated for 100–200 min (as judged by the kinetic experiments). Prism 7 (GraphPad) was used for the statistical analyses and bar-graph data representations. Heatmaps were made in RStudio using gplots and RColorBrewer libraries.

Dissociation experiments were conducted to measure any off-rates ($k_{off}$) to verify that the addition of unlabeled m$^7$G-mRNA to stop the reactions did not result in the dissociation of already-formed 48S PICs. PICs were assembled, and reactions were initiated as described above. A 20–30-fold excess of same unlabeled m$^7$G-mRNA was added before the reactions were initiated (no mRNA recruitment was observed) or after 2–20 min of incubation (depending on the $k_{app}$). Aliquots were loaded on the gel at indicated times, and percentage mRNA recruited was calculated. The data were fitted with a linear curve.

## ATPase assay

NADH-coupled ATPase assay was performed as described in (*Yourik et al., 2017*). Ded1 (100–500 nM) was added to reactions containing 1X Recon buffer, 2.5 mM phosphoenolpyruvate, 1 mM NADH, 1/250 dilution of the PK/LDH mix (pyruvate kinase (600–1000 units/mL) and lactate dehydrogenase (900–1400 units/mL)), 2 µM uncapped *RPL41A* mRNA, and reactions were initiated by addition of 5 mM ATP·Mg$^{2+}$. Reaction rates ($V_0$) were calculated from linear slope of plot of NADH oxidation over time measured as absorbance $A_{340}$. $k_{cat}$ was calculated by dividing $V_0$ by Ded1 concentration. To estimate the $K_m$ of ATP, ATP was titrated at 0–10 mM concentrations. Rates were plotted against the concentration of the ATP and fitted with Michaelis- Menten equation to calculate the $K_m$ of ATP.

## Fluorescence anisotropy assay

Fluorescent anisotropy assay was performed as described previously (*Walker et al., 2013*). Briefly, ssRNA labeled with fluorescein at the 3?-end was incubated with Ded1 concentration (0–750 nM) in the absence of any nucleotide or in the presence of 5 mM ADP·Mg$^2$ $^+$and ADPNP·Mg$^2$ $^+$in 1X Recon buffer, and fluorescent anisotropies were measured with excitation and emission wavelengths of 495 nM and 520 nM, respectively. The data were fitted with a hyperbolic equation.

## Acknowledgements

We thank Roy Parker and Eckhard Jankowsky for gifts of Ded1 expression plasmids, and Thomas Dever and Nicholas Guydosh for thoughtful comments and suggestions. This work was supported by Intramural Research Program of the National Institutes of Health (AGH and JRL). The funders had no role in study design, data collection and interpretation, or the decision to submit the work for publication.

## Additional information

### Competing interests
Alan G Hinnebusch: Reviewing editor, *eLife*. The other authors declare that no competing interests exist.

### Funding

| Funder | Author |
|--------|--------|
| National Institutes of Health | Jon R Lorsch<br>Alan G Hinnebusch |

The funders had no role in study design, data collection and interpretation, or the decision to submit the work for publication.

### Author contributions
Neha Gupta, Conceptualization, Resources, Data curation, Formal analysis, Investigation, Visualization, Methodology, Writing—original draft, Writing—review and editing; Jon R Lorsch, Conceptualization, Formal analysis, Supervision, Funding acquisition, Visualization, Project administration, Writing—review and editing; Alan G Hinnebusch, Conceptualization, Formal analysis, Supervision, Funding acquisition, Visualization, Writing—original draft, Project administration, Writing—review and editing

### Author ORCIDs
Neha Gupta http://orcid.org/0000-0002-1845-2940
Jon R Lorsch https://orcid.org/0000-0002-4521-4999
Alan G Hinnebusch http://orcid.org/0000-0002-1627-8395

### Decision letter and Author response
Decision letter https://doi.org/10.7554/eLife.38892.027
Author response https://doi.org/10.7554/eLife.38892.028

## Additional files

### Supplementary files
• Transparent reporting form
DOI: https://doi.org/10.7554/eLife.38892.025

### Data availability
All data generated or analysed during this study are included in the manuscript and supporting files. Source data files have been provided for Figures 1-6 and related figure supplements.

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
