## [Decision Letter]

Thank you for submitting your article "Yeast Ded1 promotes 48S translation pre-initiation complex assembly in an mRNA-specific and eIF4F-dependent manner" for consideration by *eLife*. Your article has been reviewed by three peer reviewers, including Nahum Sonenberg as the Reviewing Editor and Reviewer #1, and the evaluation has been overseen by James Manley as the Senior Editor.

The reviewers have discussed the reviews with one another and the Reviewing Editor has drafted this decision to help you prepare a revised submission.

Summary:

The authors use in vitro mRNA translation assays to study how yeast Ded1, eIF4A, and the eIF4G/ eIF4E complex function in 48S pre-initiation complex assembly and scanning along the mRNA 5'UTR. They report that Ded1 enhances mRNA recruitment for all the reporter mRNAs, but hyper-dependent mRNAs required higher concentrations of Ded1. They also show that the presences of secondary structure in the 5'UTR increase the dependence for Ded1 in 48S assembly. Importantly, they demonstrate that eIF4A is essential for 48S PIC assembly in all mRNAs and that Ded1 cannot replace the role eIF4A plays in 48S PIC assembly. In addition, they present evidence that the Ded1 interactions with the eIF4F complex are important in mRNA recruitment for some reporter mRNAs.

Essential revisions:

The reviewers believe that your paper is important in establishing an in vitro system to study the function of Ded1 in translation and for describing significant aspects of its mechanism of action. The work was expertly done and data support the conclusions. They raise, however, several shared concerns. They also suggested improvements to strengthen the manuscript. Consequently, we include the full text of the reviewers' comments. The major criticisms concern the discrepancies between your earlier publication of ded1 function in vivo and the results of this paper. We think that you can respond adequately to the criticisms of the reviewers by additional experimentation and clarification. We therefore invite you to submit a revised manuscript that successfully addresses these concerns.

Reviewer #1:

RNA secondary structure in the 5'UTR of mRNA largely determines translation efficiency by impeding the loading of the 43S ribosomal pre-initiation complex (PIC) onto the 5' end of mRNA and subsequent scanning. The yeast DEAD-box RNA helicase Ded1 cooperate with the canonical RNA helicase eIF4A in resolving the 5'UTR secondary structure. However, whether Ded1 and eIF4A perform distinct or overlapping functions remains unknown. This study for the first time provides a detailed biochemical analysis of the role of Ded1 in the promotion of 43S PIC attachment and scanning on the model and native mRNAs. The authors reconstituted Ded1 acceleration of 48S PIC assembly in a yeast translation initiation system containing purified components (40S ribosomal subunits, initiation factors, ATP and ^32^P-labeled mRNA). The system faithfully recapitulated increased Ded1 dependence of mRNAs that are Ded1-hyper-dependent in vivo. Other major findings are (1) Ded1 dependence of 48S PIC assembly correlated with the degree of mRNA secondary structure. (2) Ded1 stimulated both PIC attachment and scanning. (3) Ded1 better alleviated the inhibitory effects of the stem-loop structures in a cap-distal than cap-proximal location. (3) For a subset of mRNAs, Ded1 stimulation was completely dependent on eIF4G-eIF4E. In contrast, all mRNAs exhibited the requirement for eIF4A. (4) The interaction between the RNA3 domain of eIF4G and Ded1-CTD facilitated Ded1 function in mRNA recruitment to the PIC. (5) Deletion of the eIF4G RNA2 domain, which interacts with Ded1, or the N-terminal domain of Ded1, which interacts with eIF4A, conferred mRNA-specific defects in 48S PIC assembly. These studies show the functional importance of an eIF4G-eIF4E-eIF4A-Ded1 tetrameric complex in the initiation of translation of structured mRNAs.

Importantly, a human homolog of Ded1, DDX3, is known to alter the translation of specific mRNAs in tumors and in viral infections. Thus, the new insights into the molecular mechanisms of Ded1 function reported will be appealing to a broad readership of *eLife*.

1) General. It would be nice to confirm the role of the eIF4G-eIF4E-eIF4A-Ded1 ternary complex in translation initiation in a more physiological setting, such as crude yeast extract. Could they compare the inhibitory effects of eIF4A and Ded1 dominant-negative mutants on the translation of mRNAs with and without prominent secondary structure in this system?

2) It is not clear whether the ~20-fold excess of unlabeled capped mRNA is sufficient to stop the recruitment of ^32^P-labeled mRNA. A control showing inhibition of ^32^P-mRNA binding to the 43S PIC by different concentrations of unlabeled mRNA added at the beginning of the incubation should be provided.

3) The authors conclude that Dedd1 activity is dependent on the eIF4G-eIF4A complex, yet in Figure 3 dedd1 exhibits activity on several mRNAs (*SFT2, OST3-Sl*) in the absence of eIF4G-eIF4E complex. How is this explained?

*Reviewer #2:*

The authors use a reconstituted yeast translation system to assess the consequences of Ded1 on 48S PIC assembly using a small collection of hypo- and hyper-dependent mRNAs (identified from previous studies). They report that Ded1 hyper-dependent mRNAs are inherently less capable of PIC formation and more dependent on Ded1 for rapid recruitment in vitro (compared to in vivo Ded1 hypo-dependent mRNAs). The work appears well performed and in general the data support the conclusions drawn.

Is Ded1 normally saturating in vivo? If you look at the data in Figure 3—figure supplement 1, with the exception of *OST3* mRNA, eIF4A is sufficient to generate the majority of 48S PICs and adding Ded1 minimally improves this. Can the authors provide similar curves for *PMA1, FET3*, and *SFT2-M*? Yet, the authors show in Figure 1C. that RPL41A recruitment is appreciably stimulated by Ded1 in vitro. If Ded1 is not limiting in vivo, then the endpoints for most of the mRNAs would be similar.

Are the 48S PICs formed in vitro cap-dependent? How about the complexes formed in the presence of Ded1, but in the absence of eIF4E:eIF4G (Figure 3)?

In Figure S1I – The authors do not show a precursor/product relationship between the minor and major conformers (i.e., how do they know the minor species are "collapsing" down to the major species in the presence of Ded1? Has this experiment been done with eIF4F? Hence, it is difficult to know what these data are actually contributing to the overall story and the authors should consider removing it.

Subsection “Ded1 stimulation is completely dependent on eIF4G·eIF4E for a subset of mRNAs and all mRNAs require eIF4A in the presence or absence of Ded1”, first paragraph – I would urge the authors not to use only eIF4G to represent the eIF4E-eIF4G heterodimer, this will only cause confusion.

The authors generate eIF4G-ΔRNA2, eIF4G-ΔRNA3, Ded-ΔATP, and Ded-ΔCTD mutants to show that the effects of Ded1 on 48S PIC formation is eIF4G dependent. Can they validate in their system that the pairs generated abolish or severely affect such interaction. Since some of the observed effects with these mutants appear to indicate that a higher concentration of mutant is required to achieve enhancement of recruitment, why not combine pairs of mutants to see if this bears out? Also, in the Discussion, the authors write, "mRNA recruitment depends on Ded1 functioning within the eIF4E.eIF4G.eIF4A.Ded1 complex; and that eliminating any interaction between Ded1 and eIF4G or eIF4A necessitates a higher concentration of the mutant variant for efficient complex formation." Can they, at a minimum, show that the Ded1 mutants used in this study interact less efficiently with eIF4G?

Would any robust RNA helicase substitute for Ded1 in the reconstitution assay?

The sixth paragraph of the subsection “Ded1 and eIF4G mutants affect the maximal rates for recruitment differently depending on the mRNA” – of the Discussion appears quite speculative and could be condensed/removed.

Reviewer #3:

In this manuscript, the authors use in vitro mRNA recruitment assays to further elucidate how yeast Ded1, eIF4A and their interactions with the eIF4G/ eIF4E complex contribute to 48S pre-initiation complex assembly and scanning along the mRNA 5'UTR. The authors build on their previous studies selecting mRNAs considered to be hyper and hypo-dependent on Ded1 for translation to conduct their assays. They demonstrate that Ded1 enhances mRNA recruitment for all the reporter mRNAs with hyper-dependent mRNAs requiring a higher concentration of Ded1 to recruit mRNA. They go on to show that the presences of stem loops in the 5'UTR increase the dependence for Ded1 in 48S assembly. They then demonstrate that eIF4A is essential for 48S PIC assembly in all mRNAs and that Ded1 cannot replace the role eIF4A plays in 48S PIC assembly. In addition, they present evidence that the Ded1 interactions with the eIF4F complex are important in mRNA recruitment for some reporter mRNAs.

This is a well-written manuscript and was relatively easy to understand. The paper builds on a number of recent observations from various labs (including the authors' labs) concerning the roles of Ded1 and eIF4A in PIC assembly. I'm not sure the paper divulges anything particularly novel mechanistically, but it does confirm a number of observations made in vivo using an in vitro assay that directly assess the key event; 48S complex assembly.

1) In some aspects the in vitro assay does not necessarily reproduce the situation observed in vivo. First, Ded1 was shown previously by this group to be required for maintaining the translation of roughly 10% of yeast mRNAs (Sen et al., 2015), whereas all mRNAs tested in vitro show some dependence. Second eIF4A is shown here to be required for 48S PIC formation on all mRNAs tested, whereas previously reductions in eIF4A function affected the translation efficiency of very few mRNAs. These differences at least need to be commented upon and some rational explanation provided.

2) The reporter mRNAs that are used consist of the 5'UTR and the first 60nt of the CDS. However, in the previous paper they suggested that elements within the CDS and 3'UTR might engage in long range interactions to form inhibitory structures that confer dependence on Ded1. In addition, the role of Pab1 in 48S PIC recruitment would not be observed using the in vitro system described, yet as Pab1 also interacts with eIF4G, this could impact on the level of dependence on Ded1 in vivo for a particular mRNA. Again these limitations of the in vitro system should be discussed and the caveats they raise explored in the context of other literature.

3) When the stem loop in the 5'UTRs of SFT2 and PMA1 mRNA is eliminated, these mRNAs now exhibit similar k_max_ and K_1/2_ values to that of the hypo-dependent mRNAs with short 5'UTR (RPL41A and HOR7) and the reporter with an unstructured 5'UTR (-SL). However, *SFT2-M* and *PMA1-M* still have long 5'UTRs, 92nt & 239nt respectively, and presumably have some structural elements within then, so it seems very odd that they act like hypo-dependent mRNAs and not the hyper-dependent mRNAs like *OST3* and *FET3*?

4) Ideally the authors would show that at the concentrations used in these in vitro assays, eIF4G lacking the RNA3 and RNA2 domains does not interact with Ded1 where expected from previous literature.

---

## [Author Response]

Reviewer #1:[…] 1) General. It would be nice to confirm the role of the eIF4G-eIF4E-eIF4A-Ded1 ternary complex in translation initiation in a more physiological setting, such as crude yeast extract. Could they compare the inhibitory effects of eIF4A and Ded1 dominant-negative mutants on the translation of mRNAs with and without prominent secondary structure in this system?

There are published data available supporting the formation of an eIF4F-Ded1 complex in yeast extracts. Pulldown experiments from yeast cell extracts using antibodies against Ded1 revealed that Ded1 associates with eIF4E, eIF4G, and eIF4A (Senissar et al., 2014). Hilliker et al., 2011) showed that Ded1 associates with eIF4G⋅eIF4E in translation-competent yeast extracts when eIF4G⋅eIF4E was bound to a cap column. Using purified proteins, they also found that the Ded1-CTD domain interacts with the eIF4G-RNA3 domain. Confirming the reviewer’s prediction, they further demonstrated that addition of the ATPase mutant Ded1^E307A^ dominantly inhibited translation in the extracts. We agree with the reviewer that this line of experimentation is important and have now emphasized these previous studies in the Introduction (fourth and fifth paragraphs).

Moreover, in addition to showing that the dominant-negative Ded1^E307A^ mutant fails to stimulate recruitment of any of the natural mRNAs (Figure 1—figure supplement 1I), as the reviewer suggested, we have also now shown that Ded1^E307A^ is unable to accelerate recruitment of *-SL* mRNA, which has an unstructured 5′-UTR, nor that of *CP-8.1* and *CD-8.1* mRNAs, harboring stable stem-loops in their 5′-UTRs. These data were added in new Figure 2—figure supplement 1G and cited in Results (subsection “Evidence that Ded1 stimulates the PIC attachment and scanning steps of initiation”, end of second paragraph).

Additionally, we have now examined recruitment of *RPL41A* mRNA with WT Ded1 and Ded1^E307A^ present together in reactions, and found that Ded1^E307A^ interferes with WT Ded1 function, as the presence of 500 nM mutant enzyme increased the K_1/2_ value for WT Ded1 by ~4-fold (253 ± 27 nM with the mutant present vs. 60 ± 7 nM for WT Ded1 alone). We have added these new data in Figure 1—figure supplement 1J and discussed them in Results (subsection “Ded1 enhances the rate of recruitment of all natural mRNAs tested”, fourth paragraph).

2) It is not clear whether the ~20-fold excess of unlabeled capped mRNA is sufficient to stop the recruitment of ^32^P-labeled mRNA. A control showing inhibition of ^32^P-mRNA binding to the 43S PIC by different concentrations of unlabeled mRNA added at the beginning of the incubation should be provided.

We added a new figure showing this control (Figure 1—figure supplement 1A) and discuss it in Results: “The addition of ~20-fold excess non-radiolabeled mRNA in the quench was adequate to prevent further recruitment of ^32^P-labeled mRNA, and did not dissociate the pre-formed ^32^P-labeled 48S complexes on the timescale of the recruitment experiments.” In this experiment, different concentrations of unlabeled mRNA were added along with ^32^P-capped *RPL41A* mRNA and recruitment was measured over time. Unlabeled capped *RPL41A* was added along with ^32^P-labeled capped *RPL41A* mRNA (and ATP) at 10X (red), 20X (blue) or 33X (green) the amounts of labeled mRNA. The 20X addition completely inhibited detectable 48S PIC assembly on ^32^P-labeled mRNA, and a higher concentration of unlabeled *RPL41A* (~33X) did not confer further inhibition. The results for “20X – 2 min” (cyan) show that addition of 20X unlabeled *RPL41A* after 2 min of incubation with ^32^P-capped *RPL41A* to form 48S PICs containing ^32^P-mRNA did not result in loss of signal from the complex on the timescale of the kinetic assays employed in this study indicating that the labeled mRNA was stably bound once assembled into the 48S PIC and that the addition of the chase did not lead to its dissociation. This information has now been added to the figure supplement legends.

3) The authors conclude that Dedd1 activity is dependent on the eIF4G-eIF4A complex, yet in Figure 3 dedd1 exhibits activity on several mRNAs (SFT2, OST3-Sl) in the absence of eIF4G-eIF4E complex. How is this explained?

Our model that Ded1 acts as a part of a tetrameric complex with eIF4G⋅eIF4E⋅eIF4A (based on the higher K_1/2_ values of mutant eIF4G and Ded1 proteins impaired for their mutual interactions) does not preclude the idea that Ded1 can act independently of eIF4E⋅eIF4G on certain mRNAs. In fact, as the reviewer points out, Ded1 stimulates recruitment of four mRNAs (*SFT2, OST3, SFT2-M*, and *CD-8.1*) in the absence of eIF4E⋅eIF4G, indicating it can function to some extent outside of the eIF4G·eIF4E·eIF4A·Ded1 complex on these mRNAs (Figure 3, green bars). Ded1 could interact with other factors such as eIF4A, with the 40S ribosomal subunit, or with the mRNA itself to aid in the recruitment of these mRNAs without eIF4E⋅eIF4G (Gao et al., 2016; Guenther et al., 2018). This may represent an additional function of Ded1, at least on some mRNAs. However, because the maximal recruitment rates occur for all mRNAs in the presence of both eIF4E⋅eIF4G and Ded1 (Figure 3, orange bars), Ded1 likely functions within the eIF4G·eIF4E·eIF4A·Ded1 complex as well for these four mRNAs. We have added a more detailed explanation of these results in the Discussion (subsection “Ded1 and eIF4G mutants affect the maximal rates for recruitment differently depending on the mRNA”, last paragraph).

Reviewer #2:[…] Is Ded1 normally saturating in vivo? If you look at the data in Figure 3—figure supplement 1, with the exception of OST3 mRNA, eIF4A is sufficient to generate the majority of 48S PICs and adding Ded1 minimally improves this. Can the authors provide similar curves for PMA1, FET3, and SFT2-M? Yet, the authors show in Figure 1C. that RPL41A recruitment is appreciably stimulated by Ded1 in vitro. If Ded1 is not limiting in vivo, then the endpoints for most of the mRNAs would be similar.

We have added endpoint data for *PMA1* and *FET3* to Figure 3—figure supplement 1.

The reviewer is correct in noting that the endpoints are close to 90% in the absence of Ded1 for all mRNAs except *OST3* and *PMA1* (see new data added for *PMA1*); although we do not regard the effect of Ded1 in increasing the endpoints of these last two mRNA to be a minimal effect. More importantly, it should be noted that Ded1 confers large, order-of-magnitude increases in maximal rates of recruitment for the various Ded1-hyperdependent native mRNAs (Figure 1B and Figure 1—figure supplement 1H) and the two synthetic mRNAs with stem-loops (Figure 2C and Figure 2—figure supplement 1F), as well as ~2-3-fold rate increases for the Ded1-hypodependent and *-SL* mRNAs (Figures 1B and 2C), thus indicating that Ded1 accelerates 48S PIC assembly on all mRNAs in a manner that cannot be achieved with eIF4A alone. We believe that the Ded1 stimulation of rates of mRNA recruitment, and not only the higher endpoints achieved after long periods of incubation with Ded1, will be crucial in setting the overall rates of initiation (and thus protein synthesis) for most mRNAs in vivo. Consistent with this, depletion of Ded1 has global effects on translation initiation rates in vivo (Firczuk et al., 2013) as do mutations that affect the function of the factor (Chuang et al., 1997; Sen et al., 2015). Moreover, even at saturating Ded1, the maximum rates achieved for *OST3, FET3*, and the synthetic mRNAs *CP-8.1* and *CD-8.1,* remain well below those achieved for the other mRNAs examined, indicating that differences in recruitment rate among mRNAs should persist in vivo even if Ded1 is present at saturating levels for all mRNAs in cells, which of course might not be the case. We have edited Figure 3—figure supplement 1 to better highlight the increase in apparent rates that are achieved upon Ded1 addition for all mRNAs, even when Ded1 does not appreciably increase the recruitment endpoint.

Are the 48S PICs formed in vitro cap-dependent? How about the complexes formed in the presence of Ded1, but in the absence of eIF4E:eIF4G (Figure 3)?

Previous work from our lab (Mitchell et al., 2010; Yourik et al., 2017) has shown that the presence of 5′-cap blocks aberrant mRNA recruitment and imposes a requirement for eIF4E⋅eIF4G and eIF4A for maximal recruitment rate. The uncapped mRNAs can be recruited to 43S PICs (as observed by a gel-shift assay) without these factors, but the resulting complexes are aberrant and are not positioned at the start codon (as judged by toe-printing), suggesting that they are unable to either locate or stably associate with the start codon (Mitchell et al., 2010). Thus, the 5′-cap blocks recruitment in the absence of eIF4E⋅eIF4G or eIF4A, and the PICs recruited to capped mRNA in the presence of these factors are located at the start codon. In the current study, only 5′-capped mRNAs were used in all experiments, including Figure 3, to avoid the formation of these aberrant complexes, especially in the absence of eIF4E·eIF4G. We have now clarified these issues in the text (subsection “Ded1 stimulation is completely dependent on eIF4E·eIF4G for a subset of mRNAs and all mRNAs require eIF4A in the presence or absence of Ded1”, first paragraph).

In Figure S1I – The authors do not show a precursor/product relationship between the minor and major conformers (i.e., how do they know the minor species are "collapsing" down to the major species in the presence of Ded1? Has this experiment been done with eIF4F? Hence, it is difficult to know what these data are actually contributing to the overall story and the authors should consider removing it.

The reviewer makes a good point; accordingly, the data and corresponding text have been removed. We agree that these data did not contribute significantly to the conclusions of the paper.

Subsection “Ded1 stimulation is completely dependent on eIF4G·eIF4E for a subset of mRNAs and all mRNAs require eIF4A in the presence or absence of Ded1”, first paragraph – I would urge the authors not to use only eIF4G to represent the eIF4E-eIF4G heterodimer, this will only cause confusion.

We have changed eIF4G to eIF4E·eIF4G throughout, as suggested.

The authors generate eIF4G-ΔRNA2, eIF4G-ΔRNA3, Ded-ΔATP, and Ded-ΔCTD mutants to show that the effects of Ded1 on 48S PIC formation is eIF4G dependent. Can they validate in their system that the pairs generated abolish or severely affect such interaction. Since some of the observed effects with these mutants appear to indicate that a higher concentration of mutant is required to achieve enhancement of recruitment, why not combine pairs of mutants to see if this bears out? Also, in the Discussion, the authors write "mRNA recruitment depends on Ded1 functioning within the eIF4E.eIF4G.eIF4A.Ded1 complex; and that eliminating any interaction between Ded1 and eIF4G or eIF4A necessitates a higher concentration of the mutant variant for efficient complex formation." Can they, at a minimum, show that the Ded1 mutants used in this study interact less efficiently with eIF4G?

Previous work from several labs (Gao et al., 2016; Hilliker et al., 2011; Senissar et al., 2014) has established physical interactions between the Ded1-CTD and eIF4G-RNA3 domain, Ded1 and the eIF4G-RNA2 domain, Ded1-NTD and eIF4A, as well as between Ded1 and eIF4E in simplified systems consisting of purified Ded1 and eIF4G/eIF4A/eIF4E. Our data support the existence of these interactions and extend the previous work by providing strong evidence that these interactions are functionally important in mRNA recruitment, but to different extents on different mRNAs.

Based on the astute suggestion of the reviewer, to strengthen our conclusion that the pairwise interaction between the Ded1-CTD and eIF4G-RNA3 enhances Ded1 function, we have now performed mRNA recruitment assays in which the eIF4G-ΔRNA3 and Ded1-ΔCTD mutants are employed together. The prediction would be that if these two domains interact, combining their deletions would produce the same degree of recruitment defect as does either deletion alone because once the interaction is lost by deleting one of the domains, removing the other will have no additional effect. This is, in fact, the result we observe, and we have added the data as Figure 4—figure supplement 1E and the following text in the Results section: “To further verify that RNA3 of eIF4G and Ded1-CTD interact in mRNA recruitment, we combined the eIF4E·eIF4G-ΔRNA3 and Ded1-ΔCTD variants in the same assays (at the saturating concentrations determined for each alone). […] These data strongly suggest that the changes in k_max_ values conferred by eliminating either eIF4G-RNA3 or Ded1-CTD arise from loss of interaction between these two domains, because once the interaction is disrupted by removing one domain no further defect results from also removing the other.”

We thank the reviewer for suggesting this incisive experiment.

Unfortunately, because of problems with aggregation of the eIF4G-ΔRNA2 and Ded1-ΔNTD mutant proteins at the high concentrations required for saturation, we were unable to perform an analogous experiment to provide similar support for the interactions involving their deleted domains.

Would any robust RNA helicase substitute for Ded1 in the reconstitution assay?

We have not tested unrelated DEAD box RNA helicases in the mRNA recruitment assay. However, our data and those of other labs indicate that Ded1 plays a highly specific role in the process in conjunction with eIF4F. As noted above, Ded1 interacts with eIF4G, eIF4E, and eIF4A through its N- and C-terminal domains (Gao et al., 2016; Hilliker et al., 2011; Senissar et al., 2014), which are variable in sequence among different DEAD box proteins (Fairman-Williams et al., 2010). Our results indicate that efficient Ded1 function is dependent on the interactions of its NTD and CTD with eIF4G and eIF4A in an mRNA-specific manner (Figures 7A and 7B). It seems unlikely that these specific interactions would occur with other RNA helicases that contain unrelated N- and C-terminal domains. However, it is possible that the eIF4E⋅eIF4G-independent function of Ded1 observed for certain mRNAs (Figure 3C-E, G) might also be supported by another RNA helicase. It will be interesting to test this in future work.

The sixth paragraph of the subsection “Ded1 and eIF4G mutants affect the maximal rates for recruitment differently depending on the mRNA” of the Discussion appears quite speculative and could be condensed/removed.

This proposal in the Discussion has been simplified and revised to: “Since the *RPL41A* and *HOR7* mRNAs have short and less-structured 5′-UTRs, it is plausible that PIC attachment could be rate-limiting on these two mRNAs (Figure 7D (i)), whereas scanning of the 5′-UTR could be rate-limiting on the two cap-distal SL containing mRNAs, *SFT2* and *CD-8.1* (Figure 7D (ii)).”

Correspondingly, Figure 7D has also been simplified to make clear the speculative nature of this proposal.

Reviewer #3:

*[…] This is a well-written manuscript and was relatively easy to understand. The paper builds on a number of recent observations from various labs (including the authors' labs) concerning the roles of Ded1 and eIF4A in PIC assembly. I'm not sure the paper divulges anything particularly novel mechanistically, but it does confirm a number of observations made* in *vivo using an in vitro assay that directly assess the key event; 48S complex assembly.*

We would like to highlight some of the important findings of our work that provide novel mechanistic insights:

- We reconstituted the differential requirements for Ded1 displayed by different native mRNAs in vivo using a fully reconstituted system, while also demonstrating that Ded1 stimulates recruitment to some degree for every mRNA examined.

- We demonstrated for both native and synthetic mRNAs that strong Ded1 stimulation of 48S PIC assembly is dictated by the presence of defined, stable stem-loop structures in the 5′-UTRs, which impede recruitment in the presence of eIF4A alone.

- We established that eIF4A and Ded1, both essential DEAD-box RNA helicases in yeast, carry out distinct functions in mRNA recruitment, and that Ded1 cannot substitute for eIF4A on any mRNA examined.

- We provided evidence that Ded1 acts not only during scanning of the 5′-UTR (its commonly proposed role) but also enhances PIC attachment by unwinding mRNA structures near the 5′-cap.

- We provide the first evidence that the domains of Ded1 and eIF4G that stabilize a Ded1-eIF4F complex enhance 48S PIC assembly, providing crucial support for the model that Ded1 and eIF4F are functionally intertwined.

- For the first time, we show that different mRNAs exhibit differing requirements for particular Ded1-eIF4F interactions. Additionally, Ded1 acts both alone and in association with eIF4F to resolve local and global mRNA structures that impede PIC attachment or scanning to the start codon.

1) In some aspects the in vitro assay does not necessarily reproduce the situation observed in vivo. First, Ded1 was shown previously by this group to be required for maintaining the translation of roughly 10% of yeast mRNAs (Sen et al., 2015), whereas all mRNAs tested in vitro show some dependence. Second eIF4A is shown here to be required for 48S PIC formation on all mRNAs tested, whereas previously reductions in eIF4A function affected the translation efficiency of very few mRNAs. These differences at least need to be commented upon and some rational explanation provided.

The ribosome profiling experiments in our previous study were designed to identify relative translational changes between WT and mutant Ded1 or eIF4A strains. These experiments invariably yield different ribosomal footprint and total mRNA read counts due to different sequencing depths for different samples and the read counts must be normalized between samples to account for these differences. This normalization masks the global translational reductions in Ded1 and eIF4A mutants revealed by polysome profile analysis (Supplementary Figure 1, Sen et al., 2015)). Therefore, only relative translational efficiency (TE) changes, not absolute translational changes, were reported in our previous study; and the 10% of mRNAs mentioned above that were identified in a *ded1* mutant represents the group of mRNAs exhibiting greater than average TE reductions (by a factor of ≥2 with strong statistical support) – dubbed the Ded1-hyperdependent mRNAs. Those mRNAs exhibiting no change or even increased relative TE in *ded1* cells most likely exhibit reductions in absolute TEs in the mutant vs. WT cells, which are simply smaller in magnitude than the average TE change, and thus display an increase in relative TE in the mutant. Despite a comparable, strong reduction in bulk protein synthesis in the eIF4A mutant, <1% of mRNAs exhibit a significant ≥ 2-fold decrease in relative TE, indicating that the majority of mRNAs are impaired by very similar amounts in the eIF4A mutant. Thus, our current findings that representatives of the Ded1-hypodependent group of mRNAs are stimulated by Ded1 in vitro, only to a smaller extent compared to Ded1-hyperdependent mRNAs, and our finding that all mRNAs require eIF4A for recruitment in vitro, are both fully consistent with the conclusions of our previous profiling study. We have now included this interpretation of ribosome profiling data, and also explained the lack of any discrepancy between in vivo results from ribosome profiling and our current in vitro findings, in the Discussion under the heading “DEAD-box proteins Ded1 and eIF4A have complementary but distinct functions in mRNA recruitment”.

2) The reporter mRNAs that are used consist of the 5'UTR and the first 60nt of the CDS. However, in the previous paper they suggested that elements within the CDS and 3'UTR might engage in long range interactions to form inhibitory structures that confer dependence on Ded1. In addition, the role of Pab1 in 48S PIC recruitment would not be observed using the in vitro system described, yet as Pab1 also interacts with eIF4G, this could impact on the level of dependence on Ded1 in vivo for a particular mRNA. Again these limitations of the in vitro system should be discussed and the caveats they raise explored in the context of other literature.

We agree that long-range mRNA interactions could also play a role in Ded1-dependence. We also agree that PABP could influence Ded1 dependence, and Ded1’s interaction with eIF4G. However, even with reporter mRNAs comprised mostly of 5′-UTRs sequences and without PABP in our assay, we found good agreement between in vivo results obtained previously with similar reporter constructs assayed in *ded1* vs. WT cells (Sen et al., 2015) and our in vitro data regarding their degree of Ded1 dependence. Nevertheless, we accept the reviewer’s point and have added the following statements to the Discussion “mRNAs can form long-range interactions between their 5′-UTRs and coding sequences or 3′-UTRs, but because our reporter mRNAs consisted of only 5′-UTRs and the first 60 nucleotides of their coding sequences, interactions of this kind would not be recapitulated in our system. […] PABP can interact with eIF4G, and it would thus be useful in the future to explore how PABP-eIF4G interactions influence Ded1-eIF4G interactions and Ded1 functions in PIC recruitment and scanning.”

3) When the stem loop in the 5'UTRs of SFT2 and PMA1 mRNA is eliminated, these mRNAs now exhibit similar k_max_ and K_1/2_ values to that of the hypo-dependent mRNAs with short 5'UTR (RPL41A and HOR7) and the reporter with an unstructured 5'UTR (-SL). However, SFT2-M and PMA1-M still have long 5'UTRs, 92nt & 239nt respectively, and presumably have some structural elements within then, so it seems very odd that they act like hypo-dependent mRNAs and not the hyper-dependent mRNAs like OST3 and FET3?

It is likely that for the *SFT2* and *PMA1* mRNAs, the SLs are the predominant inhibitory structures imposing Ded1-dependent rate-limiting steps that require elevated Ded1 concentrations, and that other, weaker structures in these two mRNAs can be resolved at the lower concentrations of Ded1 sufficient to accelerate recruitment of Ded1-hypodependent mRNAs like *RPL41A* and *HOR7*, possibly with contributions from eIF4A.

4) Ideally the authors would show that at the concentrations used in these in vitro assays, eIF4G lacking the RNA3 and RNA2 domains does not interact with Ded1 where expected from previous literature.

As we wrote in our responses to reviewer #2, considerable published work using purified proteins has shown that removing these domains weakens or disrupts the interactions between eIF4G and Ded1. The increases in K_1/2_ values for these mutant proteins observed in our assays are fully consistent with diminished complex formation that can be rescued by mass action at elevated concentrations of the mutant variants. Moreover, in response to a suggestion by reviewer # 2, we have provided additional evidence that the eIF4G-ΔRNA3 and Ded1-ΔCTD truncations impair Ded1 function by abrogating the interaction between these two domains, as we showed that removing both domains simultaneously produces no more severe diminution of Ded1 function than does either truncation on its own, as described in new Figure 4—figure supplement 1E and new text in Results (subsection “Interaction between the RNA3 domain of eIF4G and Ded1-CTD stimulates mRNA recruitment”, fourth paragraph).